# GAP Safe Screening Rules for Sparse-Group Lasso

**Eugene Ndiaye, Olivier Fercoq, Alexandre Gramfort, Joseph Salmon**
LTCI, CNRS, Télécom ParisTech
Université Paris-Saclay
75013 Paris, France
`first.last@telecom-paristech.fr`

## Abstract

For statistical learning in high dimension, sparse regularizations have proven useful to boost both computational and statistical efficiency. In some contexts, it is natural to handle more refined structures than pure sparsity, such as for instance group sparsity. Sparse-Group Lasso has recently been introduced in the context of linear regression to enforce sparsity both at the feature and at the group level. We propose the first (provably) safe screening rules for Sparse-Group Lasso, i.e., rules that allow to discard early in the solver features/groups that are inactive at optimal solution. Thanks to efficient dual gap computations relying on the geometric properties of $\epsilon$-norm, safe screening rules for Sparse-Group Lasso lead to significant gains in term of computing time for our coordinate descent implementation.

## 1 Introduction

Sparsity is a critical property for the success of regression methods, especially in high dimension. Often, group (or block) sparsity is helpful when a known group structure needs to be enforced. This is for instance the case in multi-task learning [1] or multinomial logistic regression [5, Chapter 3]. In the multi-task setting, the group structure appears natural since one aims at jointly recovering signals whose supports are shared. In this context, sparsity and group sparsity are generally obtained by adding a regularization term to the data-fitting: $\ell_1$ norm for sparsity and $\ell_{1,2}$ norm for group sparsity.

Along with recent works on hierarchical regularization [12, 17] have focused on a specific case: the Sparse-Group Lasso. This method is the solution of a (convex) optimization program with a regularization term that is a convex combination of the two aforementioned norms, enforcing sparsity and group sparsity at the same time.

With such advanced regularizations, the computational burden can be particularly heavy in high dimension. Yet, it can be significantly reduced if one can exploit the known sparsity of the solution in the optimization. Following the seminal paper on "safe screening rules" [9], many contributions have investigated such strategies [21, 20, 3]. These so called safe screening rules compute some tests on dual feasible points to eliminate primal variables whose coefficients are guaranteed to be zero in the exact solution. Still, the computation of a dual feasible point can be challenging when the regularization is more complex than $\ell_1$ or $\ell_{1,2}$ norms. This is the case for the Sparse-Group Lasso as it is not straightforward to characterize if a dual point is feasible or not [20]. Here, we propose an efficient computation of the associated dual norm. It is all the more crucial since the naive implementation computes the Sparse-Group Lasso dual norm with a quadratic complexity w.r.t the groups dimensions.

We propose here efficient safe screening rules for the Sparse-Group Lasso that combine sequential rules (*i.e.,* rules that perform screening thanks to solutions obtained for a previously processed tuning parameter) and dynamic rules (*i.e.,* rules that perform screening as the algorithm proceeds) in a unified way. We elaborate on GAP safe rules, a strategy relying on dual gap computations introduced

for the Lasso [10] and to more general learning tasks in [15]. Note that alternative (unsafe) screening rules, for instance the "strong rules" [19], have been applied to the Lasso and its simple variants.

Our contributions are two fold here. First, we introduce the first safe screening rules for this problem, other alleged safe rules [20] for Sparse-Group Lasso were in fact not safe, as explained in detail in [15], and could lead to non-convergent implementation. Second, we link the Sparse-Group Lasso penalties to the $\epsilon$-norm in [6]. This allows to provide a new algorithm to efficiently compute the required dual norms, adapting an algorithm introduced in [7]. We incorporate our proposed GAP Safe rules in a block coordinate descent algorithm and show its practical efficiency in climate prediction tasks. Another strategy leveraging dual gap computations and active sets has recently been proposed under the name *Blitz* [13]. It could naturally benefit from our fast dual norm evaluations in this context.

**Notation**    For any integer $d \in \mathbb{N}$, we denote by $[d]$ the set $\{1, \ldots, d\}$. The standard Euclidean norm is written $\|\cdot\|$, the $\ell_1$ norm $\|\cdot\|_1$, the $\ell_\infty$ norm $\|\cdot\|_\infty$, and the transpose of a matrix $Q$ is denoted by $Q^\top$. We also denote $(t)_+ = \max(0, t)$. Our observation vector is $y \in \mathbb{R}^n$ and the design matrix $X = [X_1, \ldots, X_p] \in \mathbb{R}^{n \times p}$ has $p$ features, stored column-wise. We consider problems where the vector of parameters $\beta = (\beta_1, \ldots, \beta_p)^\top$ admits a natural group structure. A group of features is a subset $g \subset [p]$ and $n_g$ is its cardinality. The set of groups is denoted by $\mathcal{G}$ and we focus only on non-overlapping groups that form a partition of $[p]$. We denote by $\beta_g$ the vector in $\mathbb{R}^{n_g}$ which is the restriction of $\beta$ to the indexes in $g$. We write $[\beta_g]_j$ the $j$-th coordinate of $\beta_g$. We also use the notation $X_g \in \mathbb{R}^{n \times n_g}$ for the sub-matrix of $X$ assembled from the columns with indexes $j \in g$; similarly $[X_g]_j$ is the $j$-th column of $[X_g]$.

For any norm $\Omega$, $\mathcal{B}_\Omega$ refers to the corresponding unit ball, and $\mathcal{B}$ (resp. $\mathcal{B}_\infty$) stands for the Euclidean (resp. $\ell_\infty$) unit ball. The soft-thresholding operator (at level $\tau \geqslant 0$), $\mathcal{S}_\tau$, is defined for any $x \in \mathbb{R}^d$ by $[\mathcal{S}_\tau(x)]_j = \text{sign}(x_j)(|x_j| - \tau)_+$, while the group soft-thresholding (at level $\tau$) is $\mathcal{S}_\tau^{\text{gp}}(x) = (1 - \tau/\|x\|)_+ x$. Denoting $\Pi_\mathcal{C}$ the projection on a closed convex set $\mathcal{C}$, this yields $\mathcal{S}_\tau = \text{Id} - \Pi_{\tau \mathcal{B}_\infty}$. The sub-differential of a convex function $f : \mathbb{R}^d \to \mathbb{R}$ at $x$ is defined by $\partial f(x) = \{z \in \mathbb{R}^d : \forall y \in \mathbb{R}^d, f(x) - f(y) \geqslant z^\top(x - y)\}$. We recall that the sub-differential $\partial\|\cdot\|_1$ of the $\ell_1$ norm is $\text{sign}(\cdot)$, defined element-wise by $\forall j \in [d], \text{sign}(x)_j = \begin{cases} \{\text{sign}(x_j)\}, & \text{if } x_j \neq 0, \\ [-1, 1], & \text{if } x_j = 0. \end{cases}$

Note that the sub-differential $\partial\|\cdot\|$ of the Euclidean norm is $\partial\|\cdot\|(x) = \begin{cases} \{x/\|x\|\}, & \text{if } x \neq 0, \\ \mathcal{B}, & \text{if } x = 0. \end{cases}$

For any norm $\Omega$ on $\mathbb{R}^d$, $\Omega^D$ is the dual norm of $\Omega$, and is defined for any $x \in \mathbb{R}^d$ by $\Omega^D(x) = \max_{v \in \mathcal{B}_\Omega} v^\top x$, *e.g.*, $\|\cdot\|_1^D = \|\cdot\|_\infty$ and $\|\cdot\|^D = \|\cdot\|$. We only focus on the Sparse-Group Lasso norm, so we assume that $\Omega = \Omega_{\tau,w}$, where $\Omega_{\tau,w}(\beta) := \tau\|\beta\|_1 + (1 - \tau)\sum_{g \in \mathcal{G}} w_g\|\beta_g\|$, for $\tau \in [0, 1]$, $w = (w_g)_{g \in \mathcal{G}}$ with $w_g \geqslant 0$ for all $g \in \mathcal{G}$. The case where $w_g = 0$ for some $g \in \mathcal{G}$ together with $\tau = 0$ is excluded ($\Omega_{\tau,w}$ is not a norm in such a case).

## 2    Sparse-Group Lasso regression

For $\lambda > 0$ and $\tau \in [0, 1]$, the Sparse-Group Lasso estimator denoted by $\hat{\beta}^{(\lambda,\Omega)}$ is defined as a minimizer of the primal objective $P_{\lambda,\Omega}$ defined by:

$$\hat{\beta}^{(\lambda,\Omega)} \in \arg\min_{\beta \in \mathbb{R}^p} \frac{1}{2}\|y - X\beta\|^2 + \lambda\Omega(\beta) := P_{\lambda,\Omega}(\beta). \tag{1}$$

A dual formulation (see [4, Th. 3.3.5]) of (1) is given by

$$\hat{\theta}^{(\lambda,\Omega)} = \arg\max_{\theta \in \Delta_{X,\Omega}} \frac{1}{2}\|y\|^2 - \frac{\lambda^2}{2}\left\|\theta - \frac{y}{\lambda}\right\|^2 := D_\lambda(\theta), \tag{2}$$

where $\Delta_{X,\Omega} = \{\theta \in \mathbb{R}^n : \Omega^D(X^\top\theta) \leqslant 1\}$. The parameter $\lambda > 0$ controls the trade-off between data-fitting and sparsity, and $\tau$ controls the trade-off between features sparsity and group sparsity. In particular one recovers the Lasso [18] if $\tau = 1$, and the Group-Lasso [22] if $\tau = 0$.

For the primal problem, Fermat's rule (*cf.* Appendix for details) reads:

$$\lambda\hat{\theta}^{(\lambda,\Omega)} = y - X\hat{\beta}^{(\lambda,\Omega)} \qquad \textbf{(link-equation)} , \qquad (3)$$

$$X^{\top}\hat{\theta}^{(\lambda,\Omega)} \in \partial\Omega(\hat{\beta}^{(\lambda,\Omega)}) \qquad \textbf{(sub-differential inclusion)}. \qquad (4)$$

**Remark 1** (Dual uniqueness)**.** The dual solution $\hat{\theta}^{(\lambda,\Omega)}$ is unique, while the primal solution $\hat{\beta}^{(\lambda,\Omega)}$ might not be. Indeed, the dual formulation (2) is equivalent to $\hat{\theta}^{(\lambda,\Omega)} = \arg\min_{\theta\in\Delta_{X,\Omega}} \|\theta - y/\lambda\|$, so $\hat{\theta}^{(\lambda,\Omega)} = \Pi_{\Delta_{X,\Omega}}(y/\lambda)$ is the projection of $y/\lambda$ over the dual feasible set $\Delta_{X,\Omega}$.

**Remark 2** (Critical parameter: $\lambda_{\max}$)**.** There is a critical value $\lambda_{\max}$ such that 0 is a primal solution of (1) for all $\lambda \geqslant \lambda_{\max}$. Indeed, the Fermat's rule states $0 \in \arg\min_{\beta\in\mathbb{R}^p} \|y - X\beta\|^2/2 + \lambda\Omega(\beta) \Longleftrightarrow 0 \in \{X^{\top}y\} + \lambda\partial\Omega(0) \Longleftrightarrow \Omega^D(X^{\top}y) \leqslant \lambda$. Hence, the critical parameter is given by: $\lambda_{\max} := \Omega^D(X^{\top}y)$. Note that evaluating $\lambda_{\max}$ highly relies on the ability to (efficiently) compute the dual norm $\Omega^D$.

# 3 GAP safe rule for the Sparse-Group Lasso

The safe rule we propose here is an extension to the Sparse-Group Lasso of the GAP safe rules introduced for Lasso and Group-Lasso [10, 15]. For the Sparse-Group Lasso, the geometry of the dual feasible set $\Delta_{X,\Omega}$ is more complex (an illustration is given in Fig. 1). Hence, computing a dual feasible point is more intricate. As seen in Section 3.2, the computation of a dual feasible point strongly relies on the ability to evaluate the dual norm $\Omega^D$. This crucial evaluation is discussed in Section 4. We first detail how GAP safe screening rules can be obtained for the Sparse-Group Lasso.

## 3.1 Description of the screening rules

Safe screening rules exploit the known sparsity of the solutions of problems such as (1). They discard inactive features/groups whose coefficients are guaranteed to be zero for optimal solutions. Then, a significant reduction in computing time can be obtained ignoring "irrelevant" features/groups. The Sparse-Group Lasso benefits from two levels of screening: the safe rules can detect both group-wise zeros in the vector $\hat{\beta}^{(\lambda,\Omega)}$ and coordinate-wise zeros in the remaining groups.

To obtain useful screening rules one needs a **safe region**, *i.e.,* a set containing the optimal dual solution $\hat{\theta}^{(\lambda,\Omega)}$. Following [9], when we choose a ball $\mathcal{B}(\theta_c, r)$ with radius $r$ and centered at $\theta_c$ as a safe region, we call it a safe sphere. A safe sphere is all the more useful that $r$ is small and $\theta_c$ close to $\hat{\theta}^{(\lambda,\Omega)}$. The safe rules for the Sparse-Group Lasso work as follows: for any group $g$ in $\mathcal{G}$ and any safe sphere $\mathcal{B}(\theta_c, r)$

>***Group level safe screening rule:*** $\quad \max_{\theta\in\mathcal{B}(\theta_c,r)} \|\mathcal{S}_{\tau}(X_g^{\top}\theta)\| < (1-\tau)w_g \Rightarrow \hat{\beta}_g^{(\lambda,\Omega)} = 0,$ (5)

>***Feature level safe screening rule:*** $\quad \forall j \in g, \ \max_{\theta\in\mathcal{B}(\theta_c,r)} |X_j^{\top}\theta| < \tau \Rightarrow \hat{\beta}_j^{(\lambda,\Omega)} = 0.$ (6)

This means that provided one the last two test is true, the corresponding group or feature can be (safely) discarded. For screening variables, we rely on the following upper-bounds:

**Proposition 1.** *For all group $g \in \mathcal{G}$ and $j \in g$,*

$$\max_{\theta\in\mathcal{B}(\theta_c,r)} |X_j^{\top}\theta| \leqslant |X_j^{\top}\theta_c| + r\|X_j\|. \qquad (7)$$

*and*

$$\max_{\theta\in\mathcal{B}(\theta_c,r)} \|\mathcal{S}_{\tau}(X_g^{\top}\theta)\| \leqslant \mathcal{T}_g := \begin{cases} \|\mathcal{S}_{\tau}(X_g^{\top}\theta_c)\| + r\|X_g\| & \text{if } \|X_g^{\top}\theta_c\|_{\infty} > \tau, \\ (\|X_g^{\top}\theta_c\|_{\infty} + r\|X_g\| - \tau)_+ & \text{otherwise.} \end{cases} \qquad (8)$$

Assume now that one has found a safe sphere $\mathcal{B}(\theta_c, r)$ (their creation is deferred to Section 3.2), then the safe screening rules given by (5) and (6) read:

**Theorem 1** (Safe rules for the Sparse-Group Lasso)**.** *Using $\mathcal{T}_g$ defined in (8), we can state the following safe screening rules:*

>***Group level safe screening:*** $\qquad \forall g \in \mathcal{G}, \quad \text{if } \mathcal{T}_g < (1-\tau)w_g, \qquad \text{then } \hat{\beta}_g^{(\lambda,\Omega)} = 0,$

>***Feature level safe screening:*** $\quad \forall g \in \mathcal{G}, \forall j \in g, \quad \text{if } |X_j^{\top}\theta_c| + r\|X_j\| < \tau, \quad \text{then } \hat{\beta}_j^{(\lambda,\Omega)} = 0.$

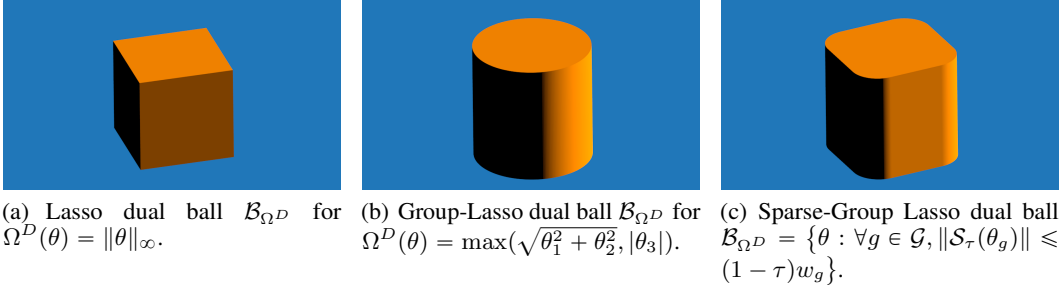

(a) Lasso dual ball $\mathcal{B}_{\Omega^D}$ for $\Omega^D(\theta) = \|\theta\|_\infty$.

(b) Group-Lasso dual ball $\mathcal{B}_{\Omega^D}$ for $\Omega^D(\theta) = \max(\sqrt{\theta_1^2 + \theta_2^2}, |\theta_3|)$.

(c) Sparse-Group Lasso dual ball $\mathcal{B}_{\Omega^D} = \{\theta : \forall g \in \mathcal{G}, \|\mathcal{S}_\tau(\theta_g)\| \leqslant (1-\tau)w_g\}$.

Figure 1: Lasso, Group-Lasso and Sparse-Group Lasso dual unit balls $\mathcal{B}_{\Omega^D} = \{\theta : \Omega^D(\theta) \leqslant 1\}$, for the case of $\mathcal{G} = \{\{1,2\},\{3\}\}$ (*i.e.*, $g_1 = \{1,2\}, g_2 = \{3\}$), $n = p = 3$, $w_{g_1} = w_{g_2} = 1$ and $\tau = 1/2$.

The screening rules can detect which coordinates or group of coordinates can be safely set to zero. This allows to remove the corresponding features from the design matrix $X$ during the optimization process. While standard algorithms solve (1) scanning all variables, only active ones, *i.e.,* non screened-out variables (using the terminology from Section 3.3) need to be considered with safe screening strategies. This leads to significant computational speed-ups, especially with a coordinate descent algorithm for which it is natural to ignore features (see Algorithm 2, in Appendix G).

## 3.2 GAP safe sphere

We now show how to compute the safe sphere radius and center using the duality gap.

### 3.2.1 Computation of the radius

With a dual feasible point $\theta \in \Delta_{X,\Omega}$ and a primal vector $\beta \in \mathbb{R}^p$ at hand, let us construct a safe sphere centered on $\theta$, with radius obtained thanks to dual gap computations.

**Theorem 2** (Safe radius). *For any $\theta \in \Delta_{X,\Omega}$ and $\beta \in \mathbb{R}^p$, one has $\hat{\theta}^{(\lambda,\Omega)} \in \mathcal{B}\left(\theta, r_{\lambda,\Omega}(\beta,\theta)\right)$, for*

$$r_{\lambda,\Omega}(\beta,\theta) = \sqrt{\frac{2(P_{\lambda,\Omega}(\beta) - D_\lambda(\theta))}{\lambda^2}},$$

i.e., *the aforementioned ball is a safe region for the Sparse-Group Lasso problem.*

*Proof.* The result holds thanks to strong concavity of the dual objective, *cf.* Appendix C. $\square$

### 3.2.2 Computation of the center

In GAP safe screening rules, the screening test relies crucially on the ability to compute a vector that belongs to the dual feasible set $\Delta_{X,\Omega}$. The geometry of this set is illustrated in Figure 1. Following [3], we leverage the primal/dual link-equation (3) to construct a dual point based on a current approximation $\beta$ of $\hat{\beta}^{(\lambda,\Omega)}$. When $\beta = \beta^{\lambda'}$ is obtained as an approximation for a previous value of $\lambda' \neq \lambda$ we call such a strategy **sequential screening**. When $\beta = \beta_k$ is the primal value at iteration $k$ obtained by an iterative algorithm, we call this **dynamical screening**. Starting from a residual $\rho = y - X\beta$, one can create a dual feasible point by choosing [1]:

$$\theta = \frac{\rho}{\max(\lambda, \Omega^D(X^\top \rho))}. \tag{9}$$

We refer to the sets $\mathcal{B}(\theta, r_{\lambda,\Omega}(\beta,\theta))$ as GAP safe spheres. Note that the generalization to any smooth data fitting term would be straightforward see [15].s

**Remark 3.** Recall that $\lambda \geqslant \lambda_{\max}$ yields $\hat{\beta}^{(\lambda,\Omega)} = 0$, in which case $\rho := y - X\hat{\beta}^{(\lambda,\Omega)} = y$ is the optimal residual and $y/\lambda_{\max}$ is the dual solution. Thus, as for getting $\lambda_{\max} = \Omega^D(X^\top y)$, the scaling computation in (9) requires a dual norm evaluation.

**Algorithm 1** Computation of $\Lambda(x, \alpha, R)$.

---

**Input:**
$x = (x_1, \ldots, x_d)^\top \in \mathbb{R}^d,\ \alpha \in [0, 1],\ R \geqslant 0$
**Output:** $\Lambda(x, \alpha, R)$

**if** $\alpha = 0$ and $R = 0$ **then**
  $\Lambda(x, \alpha, R) = \infty$
**else if** $\alpha = 0$ and $R \neq 0$ **then**
  $\Lambda(x, \alpha, R) = \|x\|/R$
**else if** $R = 0$ **then**
  $\Lambda(x, \alpha, R) = \|x\|_\infty/\alpha$
**else**
  Get $I := \left\{ i \in [d] : |x_i| > \frac{\alpha \|x\|_\infty}{\alpha + R} \right\}$
  $n_I := \mathrm{Card}(I)$
  Sort $x_{(1)} \geqslant x_{(2)} \geqslant \cdots \geqslant x_{(n_I)}$

$S_0 = x_{(0)},\ S_0^{(2)} = x_{(0)}^2,\ a_0 = 0$
**for** $k \in [n_I - 1]$ **do**
  $S_k = S_{k-1} + x_{(k)};\quad S_k^{(2)} = S_{k-1}^{(2)} + x_{(k)}^2$
  $a_{k+1} = \frac{S_k^{(2)}}{x_{(k+1)}^2} - 2\frac{S_k}{x_{(k+1)}} + k + 1$
  **if** $\frac{R^2}{\alpha^2} \in [a_k, a_{k+1}[$ **then**
    $j_0 = k + 1$
    **break**
**if** $\alpha^2 j_0 - R^2 = 0$ **then**
  $\Lambda(x, \alpha, R) = \frac{S_{j_0}^2}{2\alpha S_{j_0}}$
**else**
  $\Lambda(x, \alpha, R) = \frac{\alpha S_{j_0} - \sqrt{\alpha^2 S_{j_0}^2 - S_{j_0}^{(2)}(\alpha^2 j_0 - R^2)}}{\alpha^2 j_0 - R^2}$

---

### 3.3 Convergence of the active set

The next proposition states that the sequence of dual feasible points obtained from (9) converges to the dual solution $\hat{\theta}^{(\lambda, \Omega)}$ if $(\beta_k)_{k \in \mathbb{N}}$ converges to an optimal primal solution $\hat{\beta}^{(\lambda, \Omega)}$ (proof in Appendix). It guarantees that the GAP safe spheres $\mathcal{B}(\theta_k, r_{\lambda, \Omega}(\beta_k, \theta_k))$ are converging safe regions in the sense introduced by [10], since by strong duality $\lim_{k \to \infty} r_{\lambda, \Omega}(\beta_k, \theta_k) = 0$.

**Proposition 2.** *If* $\lim_{k \to \infty} \beta_k = \hat{\beta}^{(\lambda, \Omega)}$, *then* $\lim_{k \to \infty} \theta_k = \hat{\theta}^{(\lambda, \Omega)}$.

For any safe region $\mathcal{R}$, *i.e.*, a set containing $\hat{\theta}^{(\lambda, \Omega)}$, we define two levels of active sets, one for the group level and one for the feature level:

$$\mathcal{A}_{\mathrm{gp}}(\mathcal{R}) := \{g \in \mathcal{G},\ \max_{\theta \in \mathcal{R}} \|\mathcal{S}_\tau(X_g^\top \theta)\| \geqslant (1-\tau)w_g\},\ \mathcal{A}_{\mathrm{ft}}(\mathcal{R}) := \bigcup_{g \in \mathcal{A}_{\mathrm{gp}}(\mathcal{R})} \{j \in g :\ \max_{\theta \in \mathcal{R}} |X_j^\top \theta| \geqslant \tau\}.$$

If one considers sequence of converging regions, then the next proposition (whose proof in Appendix) states that we can identify in finite time the optimal active sets defined as follows:

$$\mathcal{E}_{\mathrm{gp}} := \left\{ g \in \mathcal{G} :\ \left\| \mathcal{S}_\tau(X_g^\top \hat{\theta}^{(\lambda, \Omega)}) \right\| = (1-\tau)w_g \right\},\ \mathcal{E}_{\mathrm{ft}} := \bigcup_{g \in \mathcal{E}_{\mathrm{gp}}} \left\{ j \in g :\ |X_j^\top \hat{\theta}^{(\lambda, \Omega)}| \geqslant \tau \right\}.$$

**Proposition 3.** *Let* $(\mathcal{R}_k)_{k \in \mathbb{N}}$ *be a sequence of safe regions whose diameters converge to 0. Then,* $\lim_{k \to \infty} \mathcal{A}_{gp}(\mathcal{R}_k) = \mathcal{E}_{gp}$ *and* $\lim_{k \to \infty} \mathcal{A}_{ft}(\mathcal{R}_k) = \mathcal{E}_{ft}$.

## 4 Properties of the Sparse-Group Lasso

To apply our safe rule, we need to be able to evaluate the dual norm $\Omega^D$ efficiently. We describe such as step hereafter along with some useful properties of the norm $\Omega$. Such evaluations are performed multiple times during the algorithm, motivating the derivation of an efficient algorithm, as presented in Algorithm 1.

### 4.1 Connections with $\epsilon$-norms

Here, we establish a link between the Sparse-Group Lasso norm $\Omega$ and the $\epsilon$-norm (denoted $\|\cdot\|_\epsilon$) introduced in [6]. For any $\epsilon \in [0, 1]$ and $x \in \mathbb{R}^d$, $\|x\|_\epsilon$ is defined as the unique nonnegative solution $\nu$ of the equation $\sum_{i=1}^d (|x_i| - (1-\epsilon)\nu)_+^2 = (\epsilon\nu)^2$, $(\|x\|_0 := \|x\|_\infty)$. Using soft-thresholding, this is equivalent to solve in $\nu$ the equation $\sum_{i=1}^d \mathcal{S}_{(1-\epsilon)\nu}(x_i)^2 = \|\mathcal{S}_{(1-\epsilon)\nu}(x)\|^2 = (\epsilon\nu)^2$. Moreover, the dual norm of the $\epsilon$-norm is given by[2]: $\|y\|_\epsilon^D = \epsilon \|y\|^D + (1-\epsilon)\|y\|_\infty^D = \epsilon \|y\| + (1-\epsilon)\|y\|_1$. Now we can express the Sparse-Group Lasso norm $\Omega$ in term of the dual $\epsilon$-norm and derive some basic properties.

**Proposition 4.** *For all groups $g$ in $\mathcal{G}$, let us introduce $\epsilon_g := \frac{(1-\tau)w_g}{\tau+(1-\tau)w_g}$. Then, the Sparse-Group Lasso norm satisfies the following properties: for any $\beta$ and $\xi$ in $\mathbb{R}^p$*

$$\Omega(\beta) = \sum_{g\in\mathcal{G}} (\tau + (1-\tau)w_g)\|\beta_g\|_{\epsilon_g}^D, \qquad and \qquad \Omega^D(\xi) = \max_{g\in\mathcal{G}} \frac{\|\xi_g\|_{\epsilon_g}}{\tau+(1-\tau)w_g}, \qquad (10)$$

$$\mathcal{B}_{\Omega^D} = \left\{\xi\in\mathbb{R}^p : \forall g\in\mathcal{G}, \|\mathcal{S}_\tau(\xi_g)\| \leqslant (1-\tau)w_g\right\}. \qquad (11)$$

*The sub-differential at $\beta$ reads $\partial\Omega(\beta) = \{z\in\mathbb{R}^p : \forall g\in\mathcal{G}, z_g \in \tau\partial\|\cdot\|_1(\beta_g) + (1-\tau)w_g\partial\|\cdot\|(\beta_g)\}$.*

We obtain from the characterization of the unit dual ball (11) that for the Sparse-Group Lasso, any dual feasible point $\theta\in\Delta_{X,\Omega}$ verifies: $\forall g\in\mathcal{G}, X_g^\top\theta \in (1-\tau)w_g\mathcal{B} + \tau\mathcal{B}_\infty$.

From the dual norm formulation (10), a vector $\theta\in\mathbb{R}^n$ is feasible if and only if $\Omega^D(X^\top\theta) \leqslant 1$, *i.e.*, $\forall g\in\mathcal{G}, \|X_g^\top\theta\|_{\epsilon_g} \leqslant \tau + (1-\tau)w_g$. Hence we deduce from (11) a new characterization of the dual feasible set: $\Delta_{X,\Omega} = \left\{\theta\in\mathbb{R}^n : \forall g\in\mathcal{G}, \|X_g^\top\theta\|_{\epsilon_g} \leqslant \tau + (1-\tau)w_g\right\}$.

### 4.2 Efficient computation of the dual norm

The following proposition shows how to compute the dual norm of the Sparse-Group Lasso (and the $\epsilon$-norm). This is turned into an efficient procedure in Algorithm 1 (see the Appendix for details).

**Proposition 5.** *For $\alpha\in[0,1], R\geqslant 0$ and $x\in\mathbb{R}^d$, the equation $\sum_{i=1}^d \mathcal{S}_{\nu\alpha}(x_i)^2 = (\nu R)^2$ has a unique solution $\nu := \Lambda(x,\alpha,R)\in\mathbb{R}_+$, that can be computed in $O(d\log d)$ operations in the worst case. With $n_I = Card\{i\in[d] : |x_i| > \alpha\|x\|_\infty/(\alpha+R)\}$, the complexity of Algorithm 1 is $n_I + n_I\log(n_I)$, which is comparable to the ambient dimension $d$.*

Thanks to Remark 2, we can explicit the critical parameter $\lambda_{\max}$ for the Sparse-Group Lasso that is

$$\lambda_{\max} = \max_{g\in\mathcal{G}} \frac{\Lambda(X_g^\top y, 1-\epsilon_g, \epsilon_g)}{\tau + (1-\tau)w_g} = \Omega^D(X^\top y), \qquad (12)$$

and get a dual feasible point (9), since $\Omega^D(X^\top\rho) = \max_{g\in\mathcal{G}} \Lambda(X_g^\top\rho, 1-\epsilon_g, \epsilon_g)/(\tau+(1-\tau)w_g)$.

## 5 Implementation

In this section we provide details on how to solve the Sparse-Group Lasso primal problem, and how we apply the GAP safe screening rules. We focus on the *block coordinate iterative soft-thresholding algorithm (ISTA-BC)*; see [16]. This algorithm requires a block-wise Lipschitz gradient condition on the data fitting term $f(\beta) = \|y - X\beta\|^2/2$. For our problem (1), one can show that for all group $g$ in $\mathcal{G}, L_g = \|X_g\|_2^2$ (where $\|\cdot\|_2$ is the spectral norm of a matrix) is a suitable block-wise Lipschitz constant. We define the block coordinate descent algorithm according to the Majorization-Minimization principle: at each iteration $l$, we choose (*e.g.*, cyclically) a group $g$ and the next iterate $\beta^{l+1}$ is defined such that $\beta_{g'}^{l+1} = \beta_{g'}^l$ if $g' \neq g$ and otherwise $\beta_g^{l+1} = \arg\min_{\beta_g\in\mathbb{R}^{n_g}} \|\beta_g - (\beta_g^l - \nabla_g f(\beta^l)/L_g)\|^2/2 + (\tau\|\beta_g\|_1 + (1-\tau)w_g\|\beta_g\|)\lambda/L_g$, where we denote for all $g$ in $\mathcal{G}, \alpha_g := \lambda/L_g$. This can be simplified to $\beta_g^{l+1} = \mathcal{S}_{(1-\tau)\omega_g\alpha_g}^{\mathrm{gp}}(\mathcal{S}_{\tau\alpha_g}(\beta_g^l - \nabla_g f(\beta^l)/L_g))$. The expensive computation of the dual gap is not performed at each pass over the data, but only every $f^{\mathrm{ce}}$ pass (in practice $f^{\mathrm{ce}} = 10$ in all our experiments). A pseudo code is given in Appendix G.

## 6 Experiments

In this section we present our experiments and illustrate the numerical benefit of screening rules for the Sparse-Group Lasso.

### 6.1 Experimental settings and methods compared

We have run our ISTA-BC algorithm [3] to obtain the Sparse-Group Lasso estimator for a non-increasing sequence of $T$ regularization parameters $(\lambda_t)_{t\in[T-1]}$ defined as follows: $\lambda_t := \lambda_{\max}10^{-\delta(t-1)/(T-1)}$.

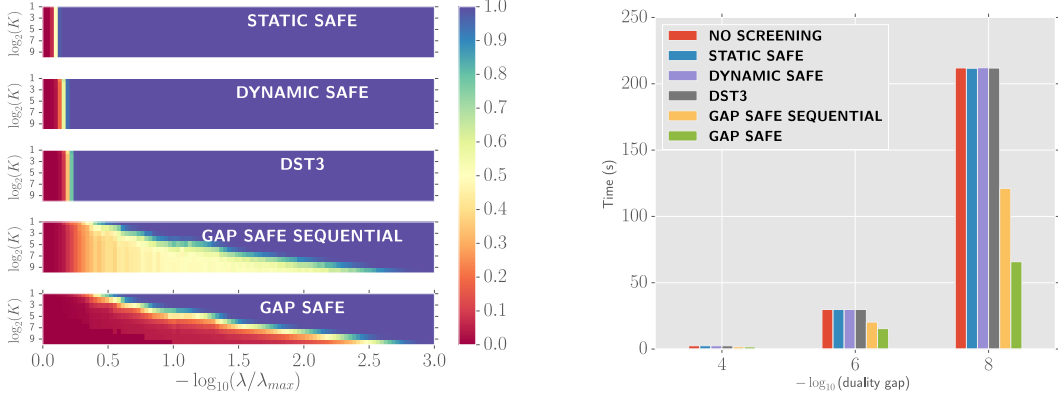

Figure 2: Experiments on a synthetic dataset ($\rho = 0.5, \gamma_1 = 10, \gamma_2 = 4, \tau = 0.2$).
(a) Proportion of active variables, *i.e.,* variables not safely eliminated, as a function of parameters $(\lambda_t)$ and the number of iterations $K$. More red, means more variables eliminated and better screening. (b) Time to reach convergence w.r.t the accuracy on the duality gap, using various screening strategies.

By default, we choose $\delta = 3$ and $T = 100$, following the standard practice when running cross-validation using sparse models (see R glmnet package [11]). The weights are always chosen as $w_g = \sqrt{n_g}$ (as in [17]).

We also provide a natural extension of the previous safe rules [9, 21, 3] to the Sparse-Group Lasso for comparisons (please refer to Appendix D for more details). The **static safe region** [9] is given by $\mathcal{B}\left(y/\lambda, \|y/\lambda_{\max} - y/\lambda\|\right)$. The corresponding **dynamic safe region** [3]) is given by $\mathcal{B}\left(y/\lambda, \|\theta_k - y/\lambda\|\right)$, where $(\theta_k)_{k \in \mathbb{N}}$ is a sequence of dual feasible points obtained by dual scaling; *cf.* Equation (9). The **DST3**, is an improvement of the preceding safe region, see [21, 3], that we adapted to the Sparse-Group Lasso. The **GAP safe sequential** rules corresponds to using only GAP Safe spheres whose centers are the (last) dual point output by the solver for a former value of $\lambda$ in the path. The **GAP safe** rules corresponds to performing our strategy both sequentially and dynamically. Presenting the sequential rule allows to measure the benefits due to sequential rules and to the dynamic rules.

We now demonstrate the efficiency of our method in both synthetic (Fig. (2)) and real datasets (Fig. 6.2). For comparison, we report computation times to reach convergence up to a certain tolerance on the duality gap for all the safe rules considered.

**Synthetic dataset:** We use a common framework [19, 20] based on the model $y = X\beta + 0.01\varepsilon$ where $\varepsilon \sim \mathcal{N}(0, \mathrm{Id}_n)$, $X \in \mathbb{R}^{n \times p}$ follows a multivariate normal distribution such that $\forall (i, j) \in [p]^2, \mathrm{corr}(X_i, X_j) = \rho^{|i-j|}$. We fix $n = 100$ and break randomly $p = 10000$ in 1000 groups of size 10 and select $\gamma_1$ groups to be active and the others are set to zero. In each selected groups, $\gamma_2$ coordinates are drawn with $[\beta_g]_j = \mathrm{sign}(\xi) \times U$ for $U$ is uniform in $[0.5, 10]$), $\xi$ uniform in $[-1, 1]$.

**Real dataset: NCEP/NCAR Reanalysis 1 [14]** The dataset contains monthly means of climate data measurements spread across the globe in a grid of $2.5° \times 2.5°$ resolutions (longitude and latitude $144 \times 73$) from 1948/1/1 to 2015/10/31 . Each grid point constitutes a group of 7 predictive variables (*Air Temperature, Precipitable water, Relative humidity, Pressure, Sea Level Pressure, Horizontal Wind Speed* and *Vertical Wind Speed*) whose concatenation across time constitutes our design matrix $X \in \mathbb{R}^{814 \times 73577}$. Such data have therefore a natural group structure.

In our experiments, we considered as target variable $y \in \mathbb{R}^{814}$, the values of *Air Temperature* in a neighborhood of Dakar. Seasonality and trend are first removed, as usually done in climate analysis for bias reduction in the regression estimates. Similar data has been used in [8], showing that the Sparse-Group Lasso estimator is well suited for prediction in climatology. Indeed, thanks to the sparsity structure, the estimates delineate via their support some predictive regions at the group level, as well as predictive features via coordinate-wise screening.

We choose $\tau$ in the set $\{0, 0.1, \ldots, 0.9, 1\}$ by splitting in $50\%$ the observations and run a training-test validation procedure. For each value of $\tau$, we require a duality gap of $10^{-8}$ on the training part

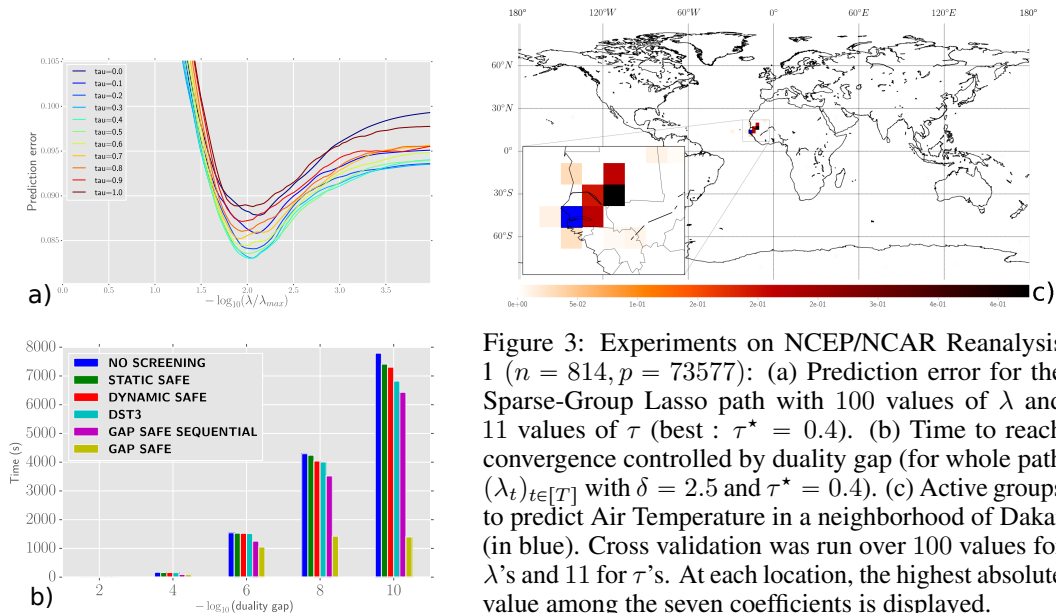

Figure 3: Experiments on NCEP/NCAR Reanalysis 1 ($n = 814, p = 73577$): (a) Prediction error for the Sparse-Group Lasso path with $100$ values of $\lambda$ and $11$ values of $\tau$ (best : $\tau^\star = 0.4$). (b) Time to reach convergence controlled by duality gap (for whole path $(\lambda_t)_{t \in [T]}$ with $\delta = 2.5$ and $\tau^\star = 0.4$). (c) Active groups to predict Air Temperature in a neighborhood of Dakar (in blue). Cross validation was run over $100$ values for $\lambda$'s and $11$ for $\tau$'s. At each location, the highest absolute value among the seven coefficients is displayed.

and pick the best one in term of prediction accuracy on the test part. The result is displayed in Figure 6.2.(a). We fixed $\delta = 2.5$ for the computational time benchmark in Figure 6.2.(b)

## 6.2 Performance of the screening rules

In all our experiments, we observe that our proposed GAP Safe rule outperforms the other rules in term of computation time. On Figure 2.(c), we can see that we need $65$s to reach convergence whereas others rules need up to $212$s at a precision of $10^{-8}$. A similar performance is observed on the real dataset (Figure 6.2) where we obtain up to a 5x speed up over the other rules. The key reason behind this performance gain is the convergence of the GAP Safe regions toward the dual optimal point as well as the efficient strategy to compute the screening rule. As shown in the results presented on Figure 2, our method still manages to screen out variables when $\lambda$ is small. It corresponds to low regularizations which lead to less sparse solutions but need to be explored during cross-validation.

In the climate experiments, the support map in Figure 6.2.(c) shows that the most important coefficients are distributed in the vicinity of the target region (in agreement with our intuition). Nevertheless, some active variables with small coefficients remain and cannot be screened out.

Note that we do not compare our method to the **TLFre** [20], since this sequential rule requires the exact knowledge of the dual optimal solution which is not available in practice. As a consequence, one may discard active variables which can prevent the algorithm from converging as shown in [15].

## 7 Conclusion

The recent GAP safe rules introduced have shown great improvements, for a wide range of regularized regression, in the reduction of computing time, especially in high dimension. To apply such GAP safe rules to the Sparse-Group Lasso, we have proposed a new description of the dual feasible set by establishing connections between the Sparse-Group Lasso norm and $\epsilon$-norms. This geometrical connection has helped providing an efficient algorithm to compute the dual norm and dual feasible points, bottlenecks for applying the GAP Safe rules. Extending GAP safe rules on general hierarchical regularizations, is a possible direction for future research.

**Acknowledgments:** this work was supported by the ANR THALAMEEG ANR-14-NEUC-0002-01, the NIH R01 MH106174, by ERC Starting Grant SLAB ERC-YStG-676943 and by the Chair Machine Learning for Big Data at Télécom ParisTech.

## Footnotes

[1] We have used a simpler scaling w.r.t. [2] choice's (without noticing much difference in practice): $\theta = s\rho$ where $s = \min\left[\max\left(\frac{\rho^\top y}{\lambda\|\rho\|^2}, \frac{-1}{\Omega^D(X^\top \rho)}\right), \frac{1}{\Omega^D(X^\top \rho)}\right]$.

[2]see [7, Eq. (42)] or Appendix

[3] The source code can be found in `https://github.com/EugeneNdiaye/GAPSAFE_SGL`.

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
