[Supplementary Material]

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

## A   Convex optimization reminder

We first recall the necessary tools for building screening rules, namely the Fermat's first order optimality condition (also called Fermat's rule) and the characterization of the sub-differential of a norm by means of its dual norm.

**Proposition 6** (Fermat's rule). *([3, Prop. 26.1]) For any convex function $f : \mathbb{R}^d \to \mathbb{R}$,*

$$x^\star \in \arg\min_{x \in \mathbb{R}^d} f(x) \Longleftrightarrow 0 \in \partial f(x^\star). \tag{13}$$

**Proposition 7.** *([2, Prop. 1.2]) The sub-differential of the norm $\Omega$ at $x$, denoted $\partial\Omega(x)$, is given by*

$$\begin{cases} \{z \in \mathbb{R}^d : \Omega^D(z) \leqslant 1\} = \mathcal{B}_{\Omega^D} & \text{if } x = 0, \\ \{z \in \mathbb{R}^d : \Omega^D(z) = 1 \text{ and } z^\top x = \Omega(x)\} & \text{otherwise.} \end{cases} \tag{14}$$

## B   Additional convexity and optimization tools

In what follows we will use the dot product notation for any $x, x' \in \mathbb{R}^d$ we write $\langle x, x' \rangle = x^\top x'$.

We denote by $\iota_C$ the indicator function of a set $C$ defined as

$$\iota_C : \mathbb{R}^d \to \mathbb{R}, \quad \iota_C(x) = \begin{cases} 0, & \text{if } x \in C, \\ +\infty, & \text{otherwise.} \end{cases} \tag{15}$$

We denote by $f^* : \mathbb{R}^d \to \mathbb{R}$ the Fenchel conjugate of $f$ defined for any $z \in \mathbb{R}^d$ by $f^*(z) = \sup_{w \in \mathbb{R}^d} w^\top z - f(w)$.

**Proposition 8.** *([2, Prop. 1.4]) The Fenchel conjugate of the norm $\Omega$ is given by*

$$\Omega^*(\xi) = \sup_{w \in \mathbb{R}^d} \left[ \xi^\top w - \Omega(w) \right] = \iota_{\mathcal{B}_{\Omega^D}}(\xi). \tag{16}$$

## C   Proofs

We first remind the simple properties underlying the concept of safe screening rules [11] in our Sparse-Group Lasso context.

**Proposition 0.** *The two levels of screening rules for the Sparse-Group Lasso are:*
***Feature level screening:***

$$\forall j \in g, \ |X_j^\top \hat{\theta}^{(\lambda,\Omega)}| < \tau \Longrightarrow \hat{\beta}_j^{(\lambda,\Omega)} = 0.$$

***Group level screening:***

$$\forall g \in \mathcal{G}, \ \left\| \mathcal{S}_\tau(X_g^\top \hat{\theta}^{(\lambda,\Omega)}) \right\| < (1-\tau)w_g \Longrightarrow \hat{\beta}_g^{(\lambda,\Omega)} = 0.$$

**Remark 4.** *The first rule is has a strict inequality, but can be relaxed to a non-strict one if $\tau \neq 1$.*

*Proof.* Let us consider $\hat{\beta}_g^{(\lambda,\Omega)} \neq 0$, $g \in \mathcal{G}$. Then combining the **subdifferential inclusion** (4), the subdifferential of the $\ell_2$-norm and the decomposition of any dual feasible point from Proposition 4, we obtain :

$$X_g^\top \hat{\theta}^{(\lambda,\Omega)} = \tau v_g + (1-\tau)w_g \frac{\hat{\beta}_g^{(\lambda,\Omega)}}{\left\| \hat{\beta}^{(\lambda,\Omega)} \right\|} \text{ where } v \in \partial \left\| \cdot \right\|_1 (\hat{\beta}^{(\lambda,\Omega)}),$$

$$X_g^\top \hat{\theta}^{(\lambda,\Omega)} = \Pi_{\tau\mathcal{B}_\infty}(X_g^\top \hat{\theta}^{(\lambda,\Omega)}) + \mathcal{S}_\tau(X_g^\top \hat{\theta}^{(\lambda,\Omega)}).$$

So we can deduce that $\mathcal{S}_\tau(X_g^\top \hat{\theta}^{(\lambda,\Omega)}) \in (1-\tau)w_g \left\{ \frac{\hat{\beta}_g^{(\lambda,\Omega)}}{\|\hat{\beta}_g^{(\lambda,\Omega)}\|} \right\}$. Since $\hat{\theta}^{(\lambda,\Omega)}$ is feasible then $\|\mathcal{S}_\tau(X_g^\top \hat{\theta}^{(\lambda,\Omega)})\| < (1-\tau)w_g$ is equivalent to $\|\mathcal{S}_\tau(X_g^\top \hat{\theta}^{(\lambda,\Omega)})\| \neq (1-\tau)w_g$ which implies, by

(a) $\mathcal{B}(\xi_c, \tilde{r}) \cap \tau\mathcal{B}_\infty \neq \varnothing$; $\xi_c \in \tau\mathring{\mathcal{B}}_\infty$

(b) $\mathcal{B}(\xi_c, \tilde{r}) \subset \tau\mathcal{B}_\infty$

(c) $\mathcal{B}(\xi_c, \tilde{r}) \cap \tau\mathcal{B}_\infty = \varnothing$; $\xi_c \notin \tau\mathring{\mathcal{B}}_\infty$

contrapositive, that $\hat{\beta}_g^{(\lambda,\Omega)} = 0$. Hence we obtain the group level safe rule. Furthermore, from the subdifferential of the $\ell_1$-norm, we have:

$$\forall j \in g, \; X_j^\top \hat{\theta}^{(\lambda,\Omega)} \in \begin{cases} (1-\tau)w_g \left\{ \dfrac{\hat{\beta}_j^{(\lambda,\Omega)}}{\|\hat{\beta}^{(\lambda,\Omega)}\|} \right\} + \tau \left\{ \text{sign}(\hat{\beta}_j^{(\lambda,\Omega)}) \right\}, & \text{if } \hat{\beta}_j^{(\lambda,\Omega)} \neq 0, \\ [-\tau, \tau], & \text{if } \hat{\beta}_j^{(\lambda,\Omega)} = 0. \end{cases}$$

Hence, if $\hat{\beta}_j^{(\lambda,\Omega)} \neq 0$ then $X_j^\top \hat{\theta}^{(\lambda,\Omega)} = \text{sign}(\hat{\beta}_j^{(\lambda,\Omega)}) \left[ (1-\tau)w_g \dfrac{|\hat{\beta}_j^{(\lambda,\Omega)}|}{\|\hat{\beta}^{(\lambda,\Omega)}\|} + \tau \right]$ and so $|X_j^\top \hat{\theta}^{(\lambda,\Omega)}| \geqslant \tau$. By contrapositive, we obtain the feature level safe rule. $\square$

**Proposition 1.** *For all group $g \in \mathcal{G}$ and $j \in g$,*

$$\max_{\theta \in \mathcal{B}(\theta_c, r)} |X_j^\top \theta| \leqslant |X_j^\top \theta_c| + r \, \|X_j\|. \tag{17}$$

$\max_{\theta \in \mathcal{B}(\theta_c, r)} \|\mathcal{S}_\tau(X_g^\top \theta)\|$ *is upper bounded by*

$$\mathcal{T}_g = \begin{cases} \left\| \mathcal{S}_\tau(X_g^\top \theta_c) \right\| + r \, \|X_g\| & \text{if } \left\| X_g^\top \theta_c \right\|_\infty > \tau, \\ \left( \left\| X_g^\top \theta_c \right\|_\infty + r \, \|X_g\| - \tau \right)_+ & \text{otherwise.} \end{cases} \tag{18}$$

*Proof.* $|X_j^\top \theta| \leqslant |[X_g^\top(\theta - \theta_c)]_j| + |X_j^\top \theta_c| \leqslant r\|X_j\| + |X_j^\top \theta_c|$ as soon as $\theta \in \mathcal{B}(\theta_c, r)$.

Since $\theta \in \mathcal{B}(\theta_c, r)$ implies that $X_g^\top \theta \in \mathcal{B}(X_g^\top \theta_c, r\|X_g\|)$, we have $\max_{\theta \in \mathcal{B}(\theta_c, r)} \|\mathcal{S}_\tau(X_g^\top \theta)\| \leqslant \max_{\xi \in \mathcal{B}(\xi_c, \tilde{r})} \|\mathcal{S}_\tau(\xi)\|$ where $\xi_c = X_g^\top \theta_c$ and $\tilde{r} = r\|X_j\|$. From now, we just have to show how to compute $\max_{\xi \in \mathcal{B}(\xi_c, \tilde{r})} \|\mathcal{S}_\tau(\xi)\|$.

- In the case where $\xi_c \in \tau\mathring{\mathcal{B}}_\infty$, if $\|\xi_c\|_\infty + \tilde{r} \leqslant \tau$ (*i.e.,* $\mathcal{B}(\xi_c, \tilde{r}) \subset \tau\mathcal{B}_\infty$), we have $\Pi_{\tau\mathcal{B}_\infty}(\xi) = \xi$ and thus, $\max_{\xi \in \mathcal{B}(\xi_c, \tilde{r})} \|\mathcal{S}_\tau(\xi)\| = \max_{\xi \in \mathcal{B}(\xi_c, \tilde{r})} \|\xi - \Pi_{\tau\mathcal{B}_\infty}(\xi)\| = 0$.

- Otherwise if $\xi_c \in \tau\mathring{\mathcal{B}}_\infty$ and $\|\xi_c\|_\infty + \tilde{r} > \tau$, for any vector $\xi \in \partial\mathcal{B}(\xi_c, \tilde{r}) \cap (\tau\mathcal{B}_\infty)^c$ and any vector $\tilde{\xi} \in \partial\tau\mathcal{B}_\infty \cap [\xi, \xi_c]$, $\|\xi - \Pi_{\tau\mathcal{B}_\infty}(\xi)\| \leqslant \|\xi - \tilde{\xi}\| = \tilde{r} - \|\tilde{\xi} - \xi_c\|$. Hence

$$\max_{\xi \in \mathcal{B}(\xi_c, \tilde{r})} \|\xi - \Pi_{\tau\mathcal{B}_\infty}(\xi)\| \leqslant \max_{\substack{\xi \in \partial\mathcal{B}(\xi_c, \tilde{r}) \cap (\tau\mathcal{B}_\infty)^c \\ \tilde{\xi} \in \partial\tau\mathcal{B}_\infty \cap [\xi, \xi_c]}} \tilde{r} - \left\| \tilde{\xi} - \xi_c \right\|$$

$$\leqslant \tilde{r} - \min_{\xi \in \partial\tau\mathcal{B}_\infty} \|\xi - \xi_c\|$$

$$= \tilde{r} - \tau + \|\xi_c\|_\infty.$$

This upper bound is attained. Indeed, $\max_{\theta \in \mathcal{B}(\xi_c, \tilde{r})} \|\xi - \Pi_{\tau\mathcal{B}_\infty}(\xi)\| = \tilde{r} - \|\Pi_{\tau\mathcal{B}_\infty}(\hat{\xi}) - \xi_c\| = \tilde{r} - \tau + \|\xi_c\|_\infty$ where $\hat{\xi}$ is a vector in $\partial\mathcal{B}(\xi_c, \tilde{r})$ such that $\Pi_{\tau\mathcal{B}_\infty}(\hat{\xi}) = \xi_c + e_{j^\star}(\tau - \|\xi_c\|_\infty)$ and $j^\star \in \arg\max_{j \in [p]} |(\xi_c)_j|$.

- If $\xi_c \notin \tau \mathring{\mathcal{B}}_\infty$, since the projection operator on a convex set is a contraction, we have

$$\forall \xi \in \partial \mathcal{B}(\xi_c, \tilde{r}), \; \|\xi - \Pi_{\tau \mathcal{B}_\infty}(\xi)\| \leqslant \|\xi - \Pi_{\tau \mathcal{B}_\infty}(\xi_c)\|$$
$$\leqslant \|\xi_c - \Pi_{\tau \mathcal{B}_\infty}(\xi_c)\| + \|\xi - \xi_c\|$$
$$= \|\xi_c - \Pi_{\tau \mathcal{B}_\infty}(\xi_c)\| + \tilde{r}.$$

Moreover, it is straightforward to see that the vector $\tilde{\xi} := \tilde{\gamma}\xi_c + (1 - \tilde{\gamma})\Pi_{\tau \mathcal{B}_\infty}(\xi_c)$ where $\tilde{\gamma} = 1 + \frac{\tilde{r}}{\|\xi_c\| + \|\Pi_{\tau \mathcal{B}_\infty}(\xi_c)\|}$ belongs to $\partial \mathcal{B}(\xi_c, \tilde{r})$; it verifies $\Pi_{\tau \mathcal{B}_\infty}(\xi_c) = \Pi_{\tau \mathcal{B}_\infty}(\tilde{\xi})$ and it attains this bound. $\qquad \square$

**Theorem 2** (Safe radius). *For any $\theta \in \Delta_{X,\Omega}$ and any $\beta \in \mathbb{R}^p$, one has $\hat{\theta}^{(\lambda,\Omega)} \in \mathcal{B}\left(\theta, r_{\lambda,\Omega}(\beta,\theta)\right)$, for*

$$r_{\lambda,\Omega}(\beta, \theta) = \sqrt{\frac{2(P_{\lambda,\Omega}(\beta) - D_\lambda(\theta))}{\lambda^2}},$$

i.e., *the aforementioned ball is a safe region for the Sparse-Group Lasso problem.*

*Proof.* By weak duality, $\forall \beta \in \mathbb{R}^p, D_\lambda(\hat{\theta}^{(\lambda,\Omega)}) \leqslant P_{\lambda,\Omega}(\beta)$. Then, note that the dual objective function (1) is $\lambda^2$-strongly concave. This implies:

$$\forall (\theta, \theta') \in \Delta_{X,\Omega} \times \Delta_{X,\Omega}, \quad D_\lambda(\theta) \leqslant D_\lambda(\theta') + \nabla D_\lambda(\theta')^\top(\theta - \theta') - \frac{\lambda^2}{2}\|\theta - \theta'\|^2.$$

Moreover, since $\hat{\theta}^{(\lambda,\Omega)}$ maximizes the concave function $D_\lambda$, the following inequality holds true:

$$\forall \theta \in \Delta_{X,\Omega}, \quad \nabla D_\lambda(\hat{\theta}^{(\lambda,\Omega)})^\top(\theta - \hat{\theta}^{(\lambda,\Omega)}) \leqslant 0.$$

Hence, we have for all $\theta \in \Delta_{X,\Omega}$ and $\beta \in \mathbb{R}^p$:

$$\frac{\lambda^2}{2}\left\|\theta - \hat{\theta}^{(\lambda,\Omega)}\right\|^2 \leqslant D_\lambda(\hat{\theta}^{(\lambda,\Omega)}) - D_\lambda(\theta)$$
$$\leqslant P_{\lambda,\Omega}(\beta) - D_\lambda(\theta). \qquad \square$$

**Proposition 2.** *If $\lim_{k\to\infty} \beta_k = \hat{\beta}^{(\lambda,\Omega)}$, then $\lim_{k\to\infty} \theta_k = \hat{\theta}^{(\lambda,\Omega)}$.*

*Proof.* Let $\alpha_k = \max(\lambda, \Omega_{\tau,w}^D(X^\top \rho_k))$ and recall that $\rho_k = y - X\beta_k$. We have :

$$\left\|\theta_k - \hat{\theta}^{(\lambda,\Omega)}\right\| = \left\|\frac{1}{\alpha_k}(y - X\beta_k) - \frac{1}{\lambda}(y - X\hat{\beta}^{(\lambda,\Omega)})\right\|$$
$$= \left\|\left(\frac{1}{\alpha_k} - \frac{1}{\lambda}\right)(y - X\beta_k) - \frac{(X\hat{\beta}^{(\lambda,\Omega)} - X\beta_k)}{\lambda}\right\|$$
$$\leqslant \left|\frac{1}{\alpha_k} - \frac{1}{\lambda}\right|\|y - X\beta_k\| + \left\|\frac{X\hat{\beta}^{(\lambda,\Omega)} - X\beta_k}{\lambda}\right\|.$$

If $\beta_k \to \hat{\beta}^{(\lambda,\Omega)}$, then $\alpha_k \to \max(\lambda, \Omega_{\tau,w}^D(X^\top(y - X\hat{\beta}^{(\lambda,\Omega)}))) = \max(\lambda, \lambda\Omega_{\tau,w}^D(X^\top\hat{\theta}^{(\lambda,\Omega)})) = \lambda$ since $y - X\hat{\beta}^{(\lambda,\Omega)} = \lambda\hat{\theta}^{(\lambda,\Omega)}$ thanks to the **link-equation** (3) and since $\hat{\theta}^{(\lambda,\Omega)}$ is feasible i.e., $\Omega_{\tau,w}^D(X^\top\hat{\theta}^{(\lambda,\Omega)}) \leqslant 1$. Hence, both terms in the previous inequality converge to zero. $\qquad \square$

**Proposition 3.** *Let $(\mathcal{R}_k)_{k\in\mathbb{N}}$ be a sequence of safe regions whose diameters converge to 0. Then, $\lim_{k\to\infty} \mathcal{A}_{gp}(\mathcal{R}_k) = \mathcal{E}_{gp}$ and $\lim_{k\to\infty} \mathcal{A}_{ft}(\mathcal{R}_k) = \mathcal{E}_{ft}$.*

*Proof.* We proceed by double inclusion. First let us prove that $\exists k_0$ s.t. $\forall k \geqslant k_0, \mathcal{A}_{gp}(\mathcal{R}_k) \subset \mathcal{E}_{gp}$. Indeed, since the diameter of $\mathcal{R}_k$ converges to zero, for any $\epsilon > 0$ there exist $k_0 \in \mathbb{N}, \forall k \geqslant k_0, \forall \theta \in \mathcal{R}_k, \|\theta - \hat{\theta}^{(\lambda,\Omega)}\| \leqslant \epsilon$. The triangle inequality implies that $\forall g \notin \mathcal{E}_{gp}, \|\mathcal{S}_\tau(X_g^\top\theta)\| \leqslant$

$\|\mathcal{S}_\tau(X_g^\top \theta) - \mathcal{S}_\tau(X_g^\top \hat\theta^{(\lambda,\Omega)})\| + \|\mathcal{S}_\tau(X_g^\top \hat\theta^{(\lambda,\Omega)})\|$. Since the soft-thresholding operator is 1-Lipschitz, we have:

$$\left\|\mathcal{S}_\tau(X_g^\top\theta)\right\| \leqslant \left\|X_g(\theta - \hat\theta^{(\lambda,\Omega)})\right\| + \left\|\mathcal{S}_\tau(X_g^\top\hat\theta^{(\lambda,\Omega)})\right\| \leqslant \epsilon\|X_g\| + \left\|\mathcal{S}_\tau(X_g^\top\hat\theta^{(\lambda,\Omega)})\right\|,$$

as soon as $k \geqslant k_0$. Moreover, $\forall g \notin \mathcal{E}_{\mathrm{gp}}$,

$$\left\|\mathcal{S}_\tau(X_g^\top\theta)\right\| \leqslant \max_{g\notin\mathcal{E}_{\mathrm{gp}}}\left\|\mathcal{S}_\tau(X_g^\top\theta)\right\| \leqslant \epsilon\max_{g\notin\mathcal{E}_{\mathrm{gp}}}\|X_g\| + \max_{g\notin\mathcal{E}_{\mathrm{gp}}}\left\|\mathcal{S}_\tau(X_g^\top\hat\theta^{(\lambda,\Omega)})\right\|.$$

It suffices to choose $\epsilon$ such that

$$\epsilon\max_{g\notin\mathcal{E}_{\mathrm{gp}}}\|X_g\| + \max_{g\notin\mathcal{E}_{\mathrm{gp}}}\left\|\mathcal{S}_\tau(X_g^\top\hat\theta^{(\lambda,\Omega)})\right\| < (1-\tau)w_g,$$

that is to say $\epsilon < \frac{(1-\tau)w_g - \max_{g\notin\mathcal{E}_{\mathrm{gp}}}\|\mathcal{S}_\tau(X_g^\top\hat\theta^{(\lambda,\Omega)})\|}{\max_{g\notin\mathcal{E}_{\mathrm{gp}}}\|X_g\|}$, to remove the group $g$. For any $k \geqslant k_0$, $\mathcal{E}_{\mathrm{gp}}^c = \{g \in \mathcal{G} : \|\mathcal{S}_\tau(X_g^\top\hat\theta^{(\lambda)})\| < (1-\tau)w_g\} \subset \mathcal{A}_{\mathrm{gp}}(\mathcal{R}_k)^c$, the set of variables removed by our screening rule. This proves the first inclusion.

Now we show that $\forall k \in \mathbb{N}, \mathcal{A}_{\mathrm{gp}}(\mathcal{R}_k) \supset \mathcal{E}_{\mathrm{gp}}$. Indeed, for all $g^\star \in \mathcal{E}_{\mathrm{gp}}, \|\mathcal{S}_\tau(X_{g^\star}^T\hat\theta^{(\lambda,\Omega)})\| = (1-\tau)w_{g^\star}$. Since for all $k$ in $\mathbb{N}$, $\hat\theta^{(\lambda,\Omega)} \in \mathcal{R}_k$ then $\max_{\theta\in\mathcal{R}_k}\|\mathcal{S}_\tau(X_g^\top\theta)\| \geqslant \|\mathcal{S}_\tau(X_{g^\star}^T\hat\theta^{(\lambda,\Omega)})\| = (1-\tau)w_{g^\star}$ hence the second inclusion holds.

We have shown that $\forall k \geqslant k_0, \mathcal{A}_{\mathrm{gp}}(\mathcal{R}_k) = \mathcal{E}_{\mathrm{gp}}$ and so $\mathcal{A}_{\mathrm{ft}}(\mathcal{R}_k) \subset \bigcup_{g\in\mathcal{E}_{\mathrm{gp}}}\{j \in g : \max_{\theta\in\mathcal{R}_k}|X_j^\top\theta| \geqslant \tau\}$. Moreover, the same reasoning yields $\forall g \in \mathcal{G}, \{j \in g : \max_{\theta\in\mathcal{R}_k}|X_j^\top\theta| \geqslant \tau\} \subset \{j \in g : |X_j^\top\hat\theta^{(\lambda,\Omega)}| \geqslant \tau\}$. Hence $\forall k \geqslant k_0, \mathcal{A}_{\mathrm{ft}}(\mathcal{R}_k) \subset \mathcal{A}_{\mathrm{ft}}$. The reciprocal inclusion is straightforward. $\square$

**Proposition 4.** . *For all group $g$ in $\mathcal{G}$, let $\epsilon_g := \frac{(1-\tau)w_g}{\tau+(1-\tau)w_g}$ then the Sparse-Group Lasso norm satisfies the following properties: for any vectors $\beta$ and $\xi$ in $\mathbb{R}^p$*

$$\Omega_{\tau,w}(\beta) = \sum_{g\in\mathcal{G}}(\tau + (1-\tau)w_g)\|\beta_g\|_{\epsilon_g}^D, \tag{19}$$

$$\Omega_{\tau,w}^D(\xi) = \max_{g\in\mathcal{G}}\frac{\|\xi_g\|_{\epsilon_g}}{\tau + (1-\tau)w_g}, \tag{20}$$

$$\mathcal{B}_{\Omega_{\tau,w}^D} = \{\xi \in \mathbb{R}^p : \forall g \in \mathcal{G}, \|\mathcal{S}_\tau(\xi_g)\| \leqslant (1-\tau)w_g\}. \tag{21}$$

*The subdifferential $\partial\Omega_{\tau,w}(\beta)$ of the norm $\Omega_{\tau,w}$ at $\beta$ is given by*

$$\left\{x \in \mathbb{R}^p : \forall g \in \mathcal{G}, x_g \in \tau\partial\|\cdot\|_1(\beta_g) + (1-\tau)w_g\partial\|\cdot\|(\beta_g)\right\}.$$

*Proof.*

$$\forall\beta \in \mathbb{R}^p, \Omega(\beta) = \tau\|\beta\|_1 + (1-\tau)\sum_{g\in\mathcal{G}}w_g\|\beta_g\| = \sum_{g\in\mathcal{G}}\left(\tau\|\beta_g\|_1 + (1-\tau)w_g\|\beta_g\|\right)$$

$$= \sum_{g\in\mathcal{G}}(\tau + (1-\tau)w_g)\left[\frac{\tau}{\tau + (1-\tau)w_g}\|\beta_g\|_1 + \frac{(1-\tau)w_g}{\tau + (1-\tau)w_g}\|\beta_g\|\right]$$

$$= \sum_{g\in\mathcal{G}}(\tau + (1-\tau)w_g)\left[(1-\epsilon_g)\|\beta_g\|_1 + \epsilon_g\|\beta_g\|\right]$$

$$= \sum_{g\in\mathcal{G}}(\tau + (1-\tau)w_g)\|\beta_g\|_{\epsilon_g}^D.$$

The definition of the dual norm reads $\Omega^D(\xi) = \max_{\beta:\Omega(\beta)\leqslant 1} \beta^\top \xi$, and solving this problem yields:

$$
\begin{aligned}
\Omega^D(\xi) &= \sup_{\beta:\Omega(\beta)\leqslant 1} \langle \beta, \xi \rangle = \sup_\beta \inf_{\mu>0} \langle \beta, \sum_{g\in\mathcal{G}} \xi_g \rangle - \mu \left( \sum_{g\in\mathcal{G}} \Omega_g(\beta_g) - 1 \right) \\
&= \inf_{\mu>0} \left\{ \sum_{g\in\mathcal{G}} \sup_{\beta_g} \left[ \langle \beta_g, \xi_g \rangle - \mu\Omega_g(\beta_g) \right] + \mu \right\} \\
&= \inf_{\mu>0} \left\{ \sum_{g\in\mathcal{G}} \mu\Omega_g^* \left( \frac{\xi_g}{\mu} \right) + \mu \right\} = \inf_{\mu>0} \left\{ \sum_{g\in\mathcal{G}} \iota_{\mathcal{B}_{\Omega_g^D}} \left( \frac{\xi_g}{\mu} \right) + \mu \right\} \\
&= \inf_{\mu>0} \left\{ \max_{g\in\mathcal{G}} \iota_{\mathcal{B}_{\Omega_g^D}} \left( \frac{\xi_g}{\mu} \right) + \mu \right\} = \max_{g\in\mathcal{G}} \inf_{\mu>0} \left\{ \Omega_g^* \left( \frac{\xi_g}{\mu} \right) + \mu \right\} \\
&= \max_{g\in\mathcal{G}} \inf_{\mu>0} \sup_{\beta_g} \langle \beta_g, \frac{\xi_g}{\mu} \rangle - \Omega_g(\beta_g) + \mu \\
&= \max_{g\in\mathcal{G}} \inf_{\mu>0} \sup_{u_g} \langle u_g, \xi_g \rangle - \mu(\Omega_g(u_g) - 1) \quad (\text{ with } \mu u_g = \beta_g) \\
&= \max_{g\in\mathcal{G}} \sup_{u_g:\Omega_g(u_g)\leqslant 1} \langle u_g, \xi_g \rangle = \max_{g\in\mathcal{G}} \sup_{u_g} \langle u_g, \xi_g \rangle \quad \text{s.t. } (\tau + (1-\tau)w_g) \|u_g\|_{\epsilon_g}^D \leqslant 1 \\
&= \max_{g\in\mathcal{G}} \sup_{u_g:\Omega_g(u_g)\leqslant 1} \langle u_g, \xi_g \rangle = \max_{g\in\mathcal{G}} \sup_{u'_g:\|u'_g\|_{\epsilon_g}^D \leqslant 1} \frac{u'^\top_g \xi_g}{\tau + (1-\tau)w_g} = \max_{g\in\mathcal{G}} \frac{\|\xi_g\|_{\epsilon_g}}{\tau + (1-\tau)w_g}.
\end{aligned}
$$

We recall here the proof of [23] for the sake of completeness. First let us write $\Omega(\beta) = \Omega_1(\beta) + \Omega_2(\beta)$, where $\Omega_1(\beta) = \tau\|\beta\|_1$ and $\Omega_2(\beta) = (1-\tau)\sum_{g\in\mathcal{G}} w_g\|\beta_g\|_2$. Since $\Omega_1$ and $\Omega_2$ are continuous everywhere, we have (see [14, Theorem 1]): $\Omega^*(\xi) = (\Omega_1 + \Omega_2)^*(\xi) = \min_{a+b=\xi}[\Omega_1^*(a) + \Omega_2^*(b)] = \min_a[\Omega_1^*(a) + \Omega_2^*(\xi - a)]$, which is also the inf-convolution (see [3, Chapter 12]) of these two norms. Using the Fenchel conjugate of the $\ell_1$ norm ($\Omega_1^* = \iota_{\tau\mathcal{B}_\infty}$) and of the $\ell_2$ norm ($\Omega_2^* = \iota_\mathcal{B}$), we have

$$
\Omega^*(\xi) = \sum_{g\in\mathcal{G}} \min_{a_g} \iota_{\tau\mathcal{B}_\infty}(a_g) + \iota_\mathcal{B} \left( \frac{\xi_g - a_g}{(1-\tau)w_g} \right) = \sum_{g\in\mathcal{G}} \iota_\mathcal{B} \left( \frac{\xi_g - \Pi_{\tau\mathcal{B}_\infty}(\xi_g)}{(1-\tau)w_g} \right).
$$

Hence the indicator of the unit dual ball is $\iota_{\mathcal{B}_{\Omega^D}}(\xi) = \sum_{g\in\mathcal{G}} \iota_{(1-\tau)w_g\mathcal{B}} (\xi_g - \Pi_{\tau\mathcal{B}_\infty}(\xi_g))$ and using $\mathcal{S}_\tau(\xi_g) = \xi_g - \Pi_{\tau\mathcal{B}_\infty}$, we have:

$$
\mathcal{B}_{\Omega^D} = \{\xi \in \mathbb{R}^p : \Omega^D(\xi) \leqslant 1\} = \{\xi \in \mathbb{R}^p : \forall g \in \mathcal{G}, \|\mathcal{S}_\tau(\xi_g)\| \leqslant (1-\tau)w_g\}.
$$

$\square$

**Proposition 5.** . *For $\alpha \in [0,1]$, $R \geqslant 0$ and $x \in \mathbb{R}^d$, the equation $\sum_{j=1}^d \mathcal{S}_{\nu\alpha}(x_j)^2 = (\nu R)^2$ has a unique solution $\nu \in \mathbb{R}_+$, denoted by $\Lambda(x, \alpha, R)$ and that can be computed in $O(d \log d)$ operations in worst case. With $n_I = \mathrm{Card}\{i \in [d] : |x_i| > \alpha\|x\|_\infty/(\alpha + R)\}$, the complexity of Algorithm 1 is $n_I + n_I \log(n_I)$, which is comparable to the ambient dimension $d$.*

*Proof.* Dividing by $\nu^2$, which is positive as soon as $x \neq 0$, we get that $\sum_{j=1}^d \mathcal{S}_{\nu\alpha}(x_j)^2 = (\nu R)^2$ is equivalent to $\sum_{j=1}^d \mathcal{S}_\alpha(x_j/\nu)^2 = R^2$. Note that $\sum_{j=1}^d \mathcal{S}_\alpha(x_j/\nu)^2 = \sum_{j=1}^d \mathcal{S}_\alpha(|x_j|/\nu)^2$ so without loss of generality we assume $x \in \mathbb{R}_+^d$.

The case $\alpha = 0$ and $R = 0$ corresponds to the situation where all $x_j$ are equal to zero or we impose $\nu$ equals to infinity. So we avoid this trivial case.

If $\alpha = 0$ and $R \neq 0$, $\nu = \|x\|/R$. Indeed,

$$
\sum_{j=1}^d \mathcal{S}_0(x_j/\nu)^2 = R^2 \iff \sum_{j=1}^d (x_j/\nu)^2 = R^2 \iff \frac{\|x\|_2^2}{\nu^2} = R^2 \text{ hence the result.}
$$

If $\alpha \neq 0$ and $R = 0$, we have :

$$\sum_{j=1}^{d} \mathcal{S}_\alpha \left( \frac{x_j}{\nu} \right)^2 = 0 \iff \forall j \in [d], \left( \frac{x_j}{\nu} - \alpha \right)_+ = 0 \iff \forall j \in [d], \frac{x_j}{\nu} \leqslant \alpha \iff \nu \geqslant \frac{\max_{j\in[d]} x_j}{\alpha}.$$

So we choose the smallest $\nu$ *i.e.*, $\nu = \|x\|_\infty / \alpha$. In all the above cases, the computation is done in $O(d)$.

Otherwise $\alpha \neq 0$ and $R \neq 0$. The function $\nu \mapsto \sum_{j=1}^{d} \mathcal{S}_\alpha (x_j/\nu)^2$ is a non-increasing continuous function with limit $+\infty$ (resp. 0) when $\nu \to 0$ (resp. $\nu \to +\infty$). Hence, there is a unique solution to $\sum_{j=1}^{d} \mathcal{S}_\alpha (x_j/\nu)^2 = R^2$.

We denote by $x_{(1)}, \ldots, x_{(d)}$ the coordinates of $x$ ordered in decreasing order (with the convention $x_{(0)} = +\infty$ and $x_{(d+1)} = 0$). Note that $\sum_{j=1}^{d} \mathcal{S}_\alpha(x_j/\nu)^2 = \sum_{j=1}^{d} \mathcal{S}_\alpha(x_{(j)}/\nu)^2$. Then, there exists an index $j_0 \in [p]$ such that

$$R^2 \in \left[ \sum_{j=0}^{d} \mathcal{S}_\alpha \left( \alpha \frac{x_{(j)}}{x_{(j_0)}} \right)^2, \sum_{j=0}^{d} \mathcal{S}_\alpha \left( \alpha \frac{x_{(j)}}{x_{(j_0+1)}} \right)^2 \right). \tag{22}$$

For such a $j_0$, one can check that $\nu \in (x_{(j_0+1)}/\alpha, x_{(j_0)}/\alpha]$. The definition of the soft-thresholding operator yields

$$\mathcal{S}_\alpha(x_j/\nu)^2 = \begin{cases} (x_j/\nu - \alpha)^2 & \text{if } x_j \geqslant \nu\alpha, \\ 0 & \text{if } x_j < \nu\alpha. \end{cases} \tag{23}$$

It can be simplified thanks to $x_j \geqslant x_{(j_0)} \Rightarrow x_j \geqslant \nu\alpha$ and $x_j \leqslant x_{(j_0+1)} \Rightarrow x_j < \nu\alpha$. Hence, $R^2 = \sum_{j=1}^{j_0} (x_{(j)}/\nu - \alpha)^2 = \sum_{j=1}^{j_0} (x_{(j)}/\nu)^2 + \alpha^2 \sum_{j=1}^{j_0} 1 - 2\alpha \sum_{j=1}^{j_0} x_{(j)}/\nu$ so solving $\sum_{j=1}^{p} \mathcal{S}_\alpha(x_{(j)}/\nu)^2 = R^2$ is equivalent to solve on $\mathbb{R}_+$

$$(\alpha^2 j_0 - R^2)\nu^2 - \left( 2\alpha \sum_{j=1}^{j_0} x_{(j)} \right) \nu + \sum_{j=1}^{j_0} x_{(j)}^2 = 0. \tag{24}$$

If $(\alpha^2 j_0 - R^2) = 0$, then $\nu = \sum_{j=1}^{j_0} x_{(j)}^2 / (2\alpha \sum_{j=1}^{j_0} x_{(j)})$. Otherwise $\nu$ is the unique solution lying in $(x_{(j_0+1)}/\alpha, x_{(j_0)}/\alpha]$ of the quadratic equation stated in Eq. (24).

In the worst case, to compute $\Lambda(x, \alpha, R)$, one needs to sort a vector of size $d$, what can be done in $O(d \log(d))$ operations, and finding $j_0$ thanks to (22) requires $O(d^2)$ if we apply a naive algorithm.

In the following, we show that one can easily reduce the complexity to $O(d \log(d))$ in worst case.

For all $j$ in $[d]$, $\mathcal{S}_\alpha \left( \alpha \frac{x_j}{x_{j_0}} \right) = 0$ as soon as $x_j \leqslant x_{j_0}$. This implies that (22) is equivalent to

$$R^2 \in \left[ \sum_{j=0}^{j_0-1} \mathcal{S}_\alpha \left( \alpha \frac{x_{(j)}}{x_{(j_0)}} \right)^2, \sum_{j=0}^{j_0} \mathcal{S}_\alpha \left( \alpha \frac{x_{(j)}}{x_{(j_0+1)}} \right)^2 \right). \tag{25}$$

Denoting $S_{j_0} := \sum_{j=1}^{j_0} x_{(j)}$ and $S_{j_0}^{(2)} := \sum_{j=1}^{j_0} x_{(j)}^2$, a direct calculation show that (25) can be rewritten as

$$R^2 \in \alpha^2 \left[ \frac{S_{j_0-1}^{(2)}}{x_{(j_0)}^2} - 2\frac{S_{j_0-1}}{x_{(j_0)}} + j_0, \frac{S_{j_0}^{(2)}}{x_{(j_0+1)}^2} - 2\frac{S_{j_0}}{x_{(j_0+1)}} + j_0 + 1 \right). \tag{26}$$

Finally, solving $\sum_{j=1}^{p} \mathcal{S}_\alpha(x_{(j)}/\nu)^2 = R^2$ is equivalent to finding the solution of $(\alpha^2 j_0 - R^2)\nu^2 - (2\alpha S_{j_0})\nu + S_{j_0}^{(2)} = 0$ lying in $(x_{(j_0+1)}/\alpha, x_{(j_0)}/\alpha]$. Hence,

$$\Lambda(x, \alpha, R) = \frac{\alpha S_{j_0} - \sqrt{\alpha^2 S_{j_0}^2 - S_{j_0}^{(2)}(\alpha^2 j_0 - R^2)}}{\alpha^2 j_0 - R^2} =: \nu_1 \tag{27}$$

or

$$\Lambda(x, \alpha, R) = \frac{\alpha S_{j_0} + \sqrt{\alpha^2 S_{j_0}^2 - S_{j_0}^{(2)}(\alpha^2 j_0 - R^2)}}{\alpha^2 j_0 - R^2} =: \nu_2. \tag{28}$$

- If $\alpha^2 j_0 - R^2 < 0$, then $\nu_2 < 0$ and so it cannot be a solution since $\Lambda(x, \alpha, R)$ must be positive.

- Otherwise, we have

$$\nu_2 \geqslant \frac{\alpha S_{j_0}}{\alpha^2 j_0 - R^2} = \frac{1}{\alpha(j_0 - \frac{R^2}{\alpha^2})} \sum_{j=1}^{j_0} x_{(j)} > \frac{1}{\alpha j_0} \sum_{j=1}^{j_0} x_{(j)} \geqslant \frac{x_{(j_0)}}{\alpha},$$

where the second inequality results from the fact that $j_0 > j_0 - R^2/\alpha^2$. And again $\nu_2$ cannot be a solution since $\Lambda(x, \alpha, R)$ belongs to $(x_{(j_0+1)}/\alpha, x_{(j_0)}/\alpha]$.

Hence, in all cases, the solution is given by $\nu_1$.

Moreover, we can significantly reduce the cost of the sort. In fact, for all $\nu$, we have $\|\mathcal{S}_{\alpha\nu}(x)\| \geqslant \|\mathcal{S}_{\alpha\nu}(x)\|_\infty = \max_{j\in[d]}(|x_j| - \nu\alpha)_+$. Hence, $\|\mathcal{S}_{\alpha\nu}(x)\| - \nu R \geqslant \|x\|_\infty - \nu\alpha - \nu R > 0$ if and only if $\nu < \|x\|_\infty/(\alpha + R)$. Combining this with Equation (23), we take into account only the coordinates which have an absolute value greater than $\alpha\|x\|_\infty/(\alpha + R)$. Indeed, by contrapositive, if $\nu$ is a solution then $\nu \geqslant \|x\|_\infty/\alpha + R$ hence $x_j < \alpha\|x\|_\infty/\alpha + R \Rightarrow x_j < \nu\alpha \overset{(23)}{\Rightarrow} \mathcal{S}_\alpha(x_j/\nu) = 0$.

Finally, computing $\Lambda(x, \alpha, R)$ can be done by applying Algorithm 1. Note that this algorithm is similar to [9, Algorithm 4]. □

# D  Notes about others methods

## D.1  Extension of some previous methods to the Sparse-Group Lasso

### D.1.1  Extension of [11]: *static safe region*

The *static safe region* can be obtained as in [11] using the ball $\mathcal{B}\left(y/\lambda, \|y/\lambda_{\max} - y/\lambda\|\right)$.

Indeed $y/\lambda_{\max}$ is a dual feasible point. Hence the distance between $y/\lambda$ and $y/\lambda_{\max}$ is smaller than the distance between $y/\lambda$ and $\hat{\theta}^{(\lambda,\Omega)}$ since the last point is the projection of $y/\lambda$ over the (close and convex) dual feasible set $\Delta_{X,\Omega}$.

### Extension of [4]: *dynamic safe region*

The *dynamic safe region* can be obtained as in [11] using the ball $\mathcal{B}\left(y/\lambda, \|\theta_k - y/\lambda\|\right)$, where the sequence $(\theta_k)_{k\in\mathbb{N}}$ converges to the dual optimal vector $\hat{\theta}^{(\lambda,\Omega)}$.

A sequence of dual points is required to construct such a ball, and following [11] we can consider the *dual scaling* procedure. We have chosen a simple procedure here: Let $\theta_k = \rho_k / \max(\lambda, \Omega_{\tau,w}^D(X^\top \rho_k))$, where $\rho_k := y - X\beta_k$, for a primal converging sequence $\beta_k$. Hence, one can use the safe sphere $\mathcal{B}\left(y/\lambda, \|\theta_k - y/\lambda\|\right)$ with the same reasoning as for the *static safe region*.

Hence, we can easily extend the corresponding screening rules to the Sparse-Group Lasso thanks to the formulation (6) and (5).

### Extension of [4]: *DST3 safe region*

Now we show that the safe regions proposed in [24] for the Lasso and generalized in [4] to the Group-Lasso can be adapted to the Sparse-Group Lasso. For that, we define $g_\star = \arg\max_{g\in\mathcal{G}} \Omega^D(X^\top y)$,

$$\mathcal{V}_\star = \left\{\theta \in \mathbb{R}^n : \left\|X_{g_\star}^\top \theta\right\|_{\epsilon_{g_\star}} \leqslant \tau + (1-\tau)w_{g_\star}\right\} \text{ and } \mathcal{H}_\star = \left\{\theta \in \mathbb{R}^n : \langle\theta, \eta\rangle = \tau + (1-\tau)w_{g_\star}\right\}.$$

Where $\eta$ is the vector normal to $\mathcal{V}_\star$ at $y/\lambda_{\max}$ and is given by $\eta := X_{g_\star}\nabla\|\cdot\|_{\epsilon_{g_\star}}\left(X_{g_\star}^\top y/\lambda_{\max}\right)$, where $\nabla\|\cdot\|_\epsilon(x) = \mathcal{S}_{(1-\epsilon)\|x\|_\epsilon}(x)/\|\mathcal{S}_{(1-\epsilon)\|x\|_\epsilon}(x)\|_\epsilon^D$, see Lemma 5 below. Let

$$\theta_c := \frac{y}{\lambda} - \left(\frac{\frac{\langle\eta,y\rangle}{\lambda} - (\tau + (1-\tau)w_{g_\star})}{\|\eta\|^2}\right)\eta$$

be the projection of $y/\lambda$ onto the hyperplane $\mathcal{H}_\star$ and $r_{\theta_k} := \sqrt{\|y/\lambda - \theta_k\|^2 - \|y/\lambda - \theta_c\|^2}$ where $\theta_k$ is a dual feasible vector (which can be obtained by dual scaling). Then, the following proposition holds.

**Proposition 9.** *Let $\theta_c$ and $r_{\theta_k}$ defined as above. Then $\hat{\theta}^{(\lambda,\Omega)} \in \mathcal{B}(\theta_c, r_{\theta_k})$.*

*Proof.* We set $\mathcal{H}_\star^- := \left\{ \theta \in \mathbb{R}^n : \langle \theta, \eta \rangle \leqslant \tau + (1 - \tau)w_{g_\star} \right\}$ the negative half-space induced by the hyperplane $\mathcal{H}_\star$. Since $\hat{\theta}^{(\lambda,\Omega)} \in \Delta_{X,\Omega} \subset \mathcal{V}_\star \subset \mathcal{H}_\star^-$ and $\mathcal{B}\left(\frac{y}{\lambda}, \|\frac{y}{\lambda} - \theta_k\|\right)$ is a safe region, then $\hat{\theta}^{(\lambda,\Omega)} \in \mathcal{H}_\star^- \cap \mathcal{B}\left(\frac{y}{\lambda}, \|\frac{y}{\lambda} - \theta_k\|\right)$.

Moreover, for any $\theta \in \mathcal{H}_\star^- \cap \mathcal{B}\left(\frac{y}{\lambda}, \|\frac{y}{\lambda} - \theta_k\|\right)$, we have:

$$\left\|\frac{y}{\lambda} - \theta_k\right\|^2 \geqslant \left\|\frac{y}{\lambda} - \theta\right\|^2 = \left\|\left(\frac{y}{\lambda} - \theta_c\right) + (\theta_c - \theta)\right\|^2 =$$

$$\left\|\frac{y}{\lambda} - \theta_c\right\|^2 + \|\theta_c - \theta\|^2 + 2\left\langle\frac{y}{\lambda} - \theta_c, \theta_c - \theta\right\rangle.$$

Since $\theta_c = \Pi_{\mathcal{H}_\star^-}\left(\frac{y}{\lambda}\right)$ and $\mathcal{H}_\star^-$ is convex, then $\langle \theta_c - \frac{y}{\lambda}, \theta_c - \theta \rangle \leqslant 0$. Thus

$$\left\|\frac{y}{\lambda} - \theta_k\right\|^2 \geqslant \left\|\frac{y}{\lambda} - \theta_c\right\|^2 + \|\theta_c - \theta\|^2, \text{ hence } \|\theta - \theta_c\| \leqslant \sqrt{\left\|\frac{y}{\lambda} - \theta_k\right\|^2 - \left\|\frac{y}{\lambda} - \theta_c\right\|^2} =: r_{\theta_k}.$$

Which show that $\mathcal{H}_\star^- \cap \mathcal{B}\left(\frac{y}{\lambda}, \|\frac{y}{\lambda} - \theta_k\|\right) \subset \mathcal{B}(\theta_c, r_{\theta_k})$. Hence the result. $\qquad\square$

## E    Sparse-Group Lasso plus Elastic Net

The Elastic-Net estimator ([26]) can be mixed with the Sparse-Group Lasso by considering

$$\underset{\beta \in \mathbb{R}^p}{\arg\min} \frac{1}{2} \|y - X\beta\|^2 + \lambda_1 \Omega(\beta) + \frac{\lambda_2}{2} \|\beta\|^2. \tag{29}$$

By setting $\tilde{X} = \begin{pmatrix} X \\ \sqrt{\lambda_2}\,\mathrm{Id}_p \end{pmatrix} \in \mathbb{R}^{n+p,p}$ and $\tilde{y} = \begin{pmatrix} y \\ 0 \end{pmatrix} \in \mathbb{R}^{n+p}$, we can reformulate (29) as

$$\underset{\beta \in \mathbb{R}^p}{\arg\min} \frac{1}{2} \left\|\tilde{y} - \tilde{X}\beta\right\|^2 + \lambda_1 \Omega(\beta), \tag{30}$$

and we can adapt our GAP safe rule framework to this case.

## F    Properties of the $\epsilon$-norm

We describe, for completeness, some properties of the $\epsilon$-norm. The following material is inspired from [9].

**Lemma 1.** *For all $\xi \in \mathbb{R}^d$, the $\epsilon$-decomposition reads: $\xi = \xi^\epsilon + \xi^{1-\epsilon}$, where $\xi^\epsilon := \mathcal{S}_{(1-\epsilon)\|\xi\|_\epsilon}(\xi)$ and $\xi^{1-\epsilon} := \xi - \xi^\epsilon$. Moreover, $\|\xi^\epsilon\| = \epsilon \|\xi\|_\epsilon$ and $\|\xi^{1-\epsilon}\|_\infty = (1-\epsilon)\|\xi\|_\epsilon$. Hence, the following decomposition holds for the $\epsilon$-norm: $\|\xi\|_\epsilon = \|\xi^\epsilon\| + \|\xi^{1-\epsilon}\|_\infty$.*

*Proof.* $\|\xi^\epsilon\| = \|\mathcal{S}_{(1-\epsilon)\|\xi\|_\epsilon}(\xi)\| = \epsilon\|\xi\|_\epsilon$ by definition of the $\epsilon$-norm $\|\xi\|_\epsilon$. Moreover,

$$\xi^{1-\epsilon} = \sum_{i=1}^d \left[\xi_i - \mathrm{sign}(\xi_i)(|\xi_i| - (1-\epsilon)\|\xi\|_\epsilon)_+\right] = \sum_{i=1}^d \mathrm{sign}(\xi_i)\left[|\xi_i| - (|\xi_i| - (1-\epsilon)\|\xi\|_\epsilon)_+\right].$$

Thus, using the symbol $a \vee b$ to represent $\max(a, b)$, one has

$$\|\xi^{1-\epsilon}\|_\infty = \max_{i \in [d]} |\mathrm{sign}(\xi_i)\left[|\xi_i| - (|\xi_i| - (1-\epsilon)\|\xi\|_\epsilon)_+\right]|$$

$$= \max_{\substack{i \in [d] \\ |\xi_i| \leqslant (1-\epsilon)\|\xi\|_\epsilon}} ||\xi_i| - (|\xi_i| - (1-\epsilon)\|\xi\|_\epsilon)_+| \vee \max_{\substack{i \in [d] \\ |\xi_i| > (1-\epsilon)\|\xi\|_\epsilon}} ||\xi_i| - (|\xi_i| - (1-\epsilon)\|\xi\|_\epsilon)_+|$$

$$= \max_{\substack{i \in [d] \\ |\xi_i| \leqslant (1-\epsilon)\|\xi\|_\epsilon}} |\xi_i| \vee (1-\epsilon)\|\xi\|_\epsilon = (1-\epsilon)\|\xi\|_\epsilon.$$

$\qquad\square$

**Lemma 2.** Let us define $U(\|\xi\|_\epsilon) := \{u \in \mathbb{R}^d : \|u\| \leqslant \epsilon\|\xi\|_\epsilon\}$ and $V(\|\xi\|_\epsilon) := \{v \in \mathbb{R}^d : \|v\|_\infty \leqslant (1-\epsilon)\|\xi\|_\epsilon\}$. Then

$$\xi^{(1-\epsilon)} = \underset{\substack{u \in U(\|\xi\|_\epsilon) \\ \xi = u+v}}{\arg\min} \|v\|_\infty \ \text{ and } \ \xi^\epsilon = \underset{\substack{v \in V(\|\xi\|_\epsilon) \\ \xi = u+v}}{\arg\min} \|u\| .$$

*Proof.*

• **Existence and uniqueness of the solutions**

It is clear that

$$\underset{\substack{u \in U(\|\xi\|_\epsilon) \\ \xi = u+v}}{\arg\min} \|v\|_\infty = \underset{\xi - U(\|\xi\|_\epsilon)}{\arg\min} \|v\|_\infty,$$

and

$$\underset{\substack{v \in V(\|\xi\|_\epsilon) \\ \xi = u+v}}{\arg\min} \|u\| = \underset{\xi - V(\|\xi\|_\epsilon)}{\arg\min} \|u\|.$$

Thus, these two problems have unique solution because we minimize strict convex functions onto convex sets.

• **Uniqueness of the $\epsilon$-decomposition**

From Lemma 1 we have $\xi = \xi^\epsilon + \xi^{1-\epsilon}$ where $\|\xi^\epsilon\| = \epsilon\|\xi\|_\epsilon$ and $\|\xi^{1-\epsilon}\|_\infty = (1-\epsilon)\|\xi\|_\epsilon$. Hence $\xi^\epsilon \in U(\|\xi\|_\epsilon)$ and $\xi^{1-\epsilon} \in V(\|\xi\|_\epsilon)$. Now it suffices to show that this $\epsilon$-decomposition is unique.

Suppose $\xi \neq 0$ (the uniqueness is trivial otherwise) and $v \in V(\|\xi\|_\epsilon)$. We show that for any $u \in \mathbb{R}^d$ such that $\xi = u + v$, $v \neq \xi^{1-\epsilon}$ implies $u \notin U(\|\xi\|_\epsilon)$.

$$\|u\|^2 = \|\xi - v\|^2 = \left\|\xi^\epsilon + (\xi^{1-\epsilon} - v)\right\|^2 = \|\xi^\epsilon\|^2 + 2\langle\xi^\epsilon, \xi^{1-\epsilon} - v\rangle + \left\|\xi^{1-\epsilon} - v\right\|^2,$$

hence $\|u\|^2 > \epsilon^2\|\xi\|_\epsilon^2 + 2\langle\xi^\epsilon, \xi^{1-\epsilon} - v\rangle$ because $\|\xi^\epsilon\| = \epsilon\|\xi\|_\epsilon$ and $\|\xi^{1-\epsilon} - v\| > 0$ ($v \neq \xi^{1-\epsilon}$). Moreover,

$$\langle\xi^\epsilon, \xi^{1-\epsilon} - v\rangle = \sum_{i=1}^d \left[\mathrm{sign}(\xi_i)(|\xi_i| - (1-\epsilon)\|\xi\|_\epsilon)_+\right]\left[\mathrm{sign}(\xi_i)(|\xi_i| - (|\xi_i| - (1-\epsilon)\|\xi\|_\epsilon)_+) - v_i\right]$$

$$= \sum_{i=1}^d \left[(|\xi_i| - (1-\epsilon)\|\xi\|_\epsilon)_+\right]\left[(|\xi_i| - (|\xi_i| - (1-\epsilon)\|\xi\|_\epsilon)_+) - v_i\,\mathrm{sign}(\xi_i)\right]$$

$$\geqslant \sum_{\substack{i=1 \\ |\xi_i| > (1-\epsilon)\|\xi\|_\epsilon}} \left[|\xi_i| - (1-\epsilon)\|\xi\|_\epsilon\right]\left[(1-\epsilon)\|\xi\|_\epsilon - v_i\,\mathrm{sign}(\xi_i)\right] \geqslant 0.$$

The last inequality hold because $v \in V(\|\xi\|_\epsilon)$ i.e., $\forall i \in [d]$, $v_i \leqslant (1-\epsilon)\|\xi\|_\epsilon$. Finally, $\|u\|^2 > \epsilon^2\|\xi\|_\epsilon^2$ hence the result. $\qquad\square$

**Lemma 3.** $\left\{\xi \in \mathbb{R}^d : \|\xi\|_\epsilon \leqslant \nu\right\} = \left\{u + v : u, v \in \mathbb{R}^d, \|u\| \leqslant \epsilon\nu, \|v\|_\infty \leqslant (1-\epsilon)\nu\right\}.$

*Proof.* Thanks to Lemma 1, we have $\xi = \xi^\epsilon + \xi^{1-\epsilon}$, $\|\xi^\epsilon\| = \epsilon\|\xi\|_\epsilon$ and $\|\xi^{1-\epsilon}\|_\infty = (1-\epsilon)\|\xi\|_\epsilon$. Hence, $\|\xi\|_\epsilon \leqslant \nu$ implies $\|\xi^\epsilon\| \leqslant \epsilon\nu$ and $\|\xi^{1-\epsilon}\|_\infty \leqslant (1-\epsilon)\nu$.

Suppose $\xi = u + v$ such that $\|u\| \leqslant \epsilon\nu$ and $\|v\|_\infty \leqslant (1-\epsilon)\nu$. From the $\epsilon$-decomposition, we have $\|\xi\|_\epsilon = \|\xi^\epsilon\| + \|\xi^{1-\epsilon}\|_\infty$. Moreover, $\|\xi^\epsilon\| \leqslant \|u\|$ and $\|\xi^{1-\epsilon}\|_\infty \leqslant \|v\|_\infty$ thanks to Lemma 2. Hence $\|\xi^\epsilon\| \leqslant \|u\| + \|v\|_\infty \leqslant \epsilon\nu + (1-\epsilon)\nu = \nu.$ $\qquad\square$

**Lemma 4** (Dual norm of the $\epsilon$-norm). Let $\xi \in \mathbb{R}^d$, then $\|\xi\|_\epsilon^D = \epsilon\|\xi\| + (1-\epsilon)\|\xi\|_1$.

*Proof.*

$$\|\xi\|_\epsilon^D := \max_{\|x\|_\epsilon \leqslant 1} \xi^\top x = \max_{\substack{\|u\| \leqslant \epsilon \\ \|v\|_\infty \leqslant 1-\epsilon}} \xi^\top(u+v) \text{ thanks to Lemma 3}$$

$$= \epsilon \max_{\|u\| \leqslant 1} \xi^\top u + (1-\epsilon) \max_{\|v\|_\infty \leqslant 1} \xi^\top v = \epsilon\|\xi\|^D + (1-\epsilon)\|\xi\|_\infty^D. \qquad\square$$

**Lemma 5.** Let $\xi \in \mathbb{R}^d \backslash \{0\}$. Then $\nabla \|\cdot\|_\epsilon (\xi) = \frac{\xi^\epsilon}{\|\xi^\epsilon\|_\epsilon^D}$.

*Proof.* Let us define $h : \mathbb{R} \times \mathbb{R}^d \mapsto \mathbb{R}$ by $h(\nu, \xi) = \|\mathcal{S}_{(1-\epsilon)\nu}(\xi)\| - \epsilon\nu$. Then we have

$$
\begin{aligned}
\frac{\partial h}{\partial \nu}(\nu, \xi) &= \frac{\mathcal{S}_{(1-\epsilon)\nu}(\xi)^\top}{\|\mathcal{S}_{(1-\epsilon)\nu}(\xi)\|} \frac{\partial \mathcal{S}_{(1-\epsilon)\nu}(\xi)}{\partial \nu} - \epsilon = -\frac{\mathcal{S}_{(1-\epsilon)\nu}(\xi)^\top}{\|\mathcal{S}_{(1-\epsilon)\nu}(\xi)\|}(1-\epsilon)\operatorname{sign}(\xi) - \epsilon \\
&= -\frac{\|\mathcal{S}_{(1-\epsilon)\nu}(\xi)\|_1}{\|\mathcal{S}_{(1-\epsilon)\nu}(\xi)\|}(1-\epsilon) - \epsilon = -\frac{(1-\epsilon)\|\mathcal{S}_{(1-\epsilon)\nu}(\xi)\|_1 + \epsilon\|\mathcal{S}_{(1-\epsilon)\nu}(\xi)\|}{\|\mathcal{S}_{(1-\epsilon)\nu}(\xi)\|} \\
&= -\frac{\|\mathcal{S}_{(1-\epsilon)\nu}(\xi)\|_\epsilon^D}{\|\mathcal{S}_{(1-\epsilon)\nu}(\xi)\|} \quad \text{(thanks to Lemma 4).}
\end{aligned}
$$

By definition of the $\epsilon$-norm, $h(\|\xi\|_\epsilon, \xi) = 0$. Since $\frac{\partial h}{\partial \nu}(\|\xi\|_\epsilon, \xi) = -\frac{\|\xi^\epsilon\|_\epsilon^D}{\epsilon\|\xi\|_\epsilon} \neq 0$, we obtain by applying the Implicit Function Theorem

$$
\nabla \|\cdot\|_\epsilon (\xi) \times \frac{\partial h}{\partial \nu}(\|\xi\|_\epsilon, \xi) + \frac{\partial h}{\partial \xi}(\|\xi\|_\epsilon, \xi) = 0 \text{ hence } \nabla \|\cdot\|_\epsilon (\xi) = -\frac{\frac{\partial h}{\partial \xi}(\|\xi\|_\epsilon, \xi)}{\frac{\partial h}{\partial \nu}(\|\xi\|_\epsilon, \xi)}.
$$

Moreover, $\frac{\partial h}{\partial \xi}(\|\xi\|_\epsilon, \xi) = \frac{\mathcal{S}_{(1-\epsilon)\|\xi\|_\epsilon}(\xi)}{\|\mathcal{S}_{(1-\epsilon)\|\xi\|_\epsilon}(\xi)\|} = \frac{\xi^\epsilon}{\|\xi^\epsilon\|} = \frac{\xi^\epsilon}{\epsilon\|\xi\|_\epsilon}$ hence the result: $\nabla \|\cdot\|_\epsilon (\xi) = \frac{\xi^\epsilon}{\|\xi^\epsilon\|_\epsilon^D}$. $\qquad\square$

## G   Implementation

Here we present the ISTA-BC we considered, and provide the GAP safe rules we have implemented. Note that the GAP safe rules we have used are both sequential and dynamical by nature and simply refer to as GAP safe.

---
**Algorithm 2** ISTA-BC with GAP safe rules
---
**Input**  $: X, y, \epsilon, K, f^{\mathrm{ce}}, (\lambda_t)_{t \in [T-1]}$
$\forall g \in \mathcal{G}$, compute $L_g = \|X_g\|_2^2$
Compute $\lambda_0 = \lambda_{\max}$ thanks to (12) and Algorithm 1
$\beta^{\lambda_0} = 0$
**for** $t \in [T-1]$ **do**
    $\forall g \in \mathcal{G}, \alpha_g \leftarrow \lambda_t/L_g$
    $\beta \leftarrow \beta^{\lambda_{t-1}}$                                      `// Get previous ϵ-solution`
    **for** $k \in [K]$ **do**
        **if** $k \mod f^{\mathrm{ce}} = 1$ **then**
            Compute $\theta$ thanks to (9) and Algorithm 1.
            Set $\mathcal{R} = \mathcal{B}\left(\theta, \sqrt{\frac{2(P_{\lambda_t, \Omega}(\beta) - D_{\lambda_t}(\theta))}{\lambda_t^2}}\right)$            `// Safe sphere`
            **if** $P_{\lambda_t, \Omega}(\beta) - D_{\lambda_t}(\theta) \leqslant \epsilon$             `// Stopping criterion`
            **then**
                $\beta^{\lambda_t} \leftarrow \beta$
                **break**
            Update $\mathcal{A}_{\mathrm{gp}}(\mathcal{R})$ and $\mathcal{A}_{\mathrm{ft}}(\mathcal{R})$ thanks to Theorem 2
        **for** $g \in \mathcal{A}_{gp}(\mathcal{R})$ **do**                         `// Loop over active groups`
            **for** $j \in g \cap \mathcal{A}_{ft}(\mathcal{R})$ **do**                `// Loop over active features`
                $\beta_j \leftarrow \mathcal{S}_{\tau\alpha_g}\left(\beta_j - \frac{\nabla_j f(\beta)}{L_g}\right)$     `// Component-wise Soft-thresholding step`
            $\beta_g \leftarrow \mathcal{S}^{\mathrm{gp}}_{(1-\tau)\omega_g\alpha_g}(\beta_g)$          `// Block-wise Soft-thresholding step`
**Output** $: (\beta^{\lambda_t})_{t \in [T-1]}$
---