[Reviews · NeurIPS 2016]

Reviewer 1

Summary

The paper introduces group safe and feature safe screening rules to rule out irrelevant feature (groups) in sparse group lasso, along with computation of radius and center of the resulting safe sphere. The latter involves obtaining a sequence of dual feasible points. The resulting spheres are shown to be converging safe regions and that optimal active sets can be identified in finite time. A connection is established with epsilon norms that allows for fast computation of the sparse group lasso dual norm. The approach is demonstrated on a wide range on simulated and climatology data.

Qualitative Assessment

+ The paper deal with a very relevant topic and is a pleasure to read. The exposition is clear and intuitive. The results are well discussed and convincingly demonstrated. + The second contribution of the paper, i.e., the connection to epsilon norms and its impact on computation complexity of the sparse group lasso dual norm is the most exciting part of the work. - The first contribution of the paper, namely the proposal and characterization of safe rules uses the same machinery as previous work: GAP Safe screening rules for sparse multi-task and multi-class models, NIPS 2015, e.g. radius computation and center computation are similar. This somewhat reduces the novelty of the present work.

Confidence in this Review

2-Confident (read it all; understood it all reasonably well)


Reviewer 2

Summary

This paper derives screening rules that allow to set certain variables to zero when fitting a sparse-group lasso model. The main technical contribution is a fast method for evaluating the dual norm of the regularizer. In addition, the authors describe an implementation based on coordinate descent and provide some numerical simulations.

Qualitative Assessment

This work provides a technique for accelerating the fit of a sparse-group lasso model, which relies on a novel method for evaluating the dual norm of the regularizer. This is an interesting contribution: speeding up the fit of structured sparse models is an important subject, since applying such models to large datasets is often challenging due to computational constraints. The paper motivates and explains the method clearly, provides thorough references and includes interesting numerical simulations, where the method is compared to other approaches that have been adapted to the sparse-group lasso by the authors. The only caveat is that the potential impact may be somewhat restricted, as it focuses exclusively on a rather specific sparse model.

Confidence in this Review

2-Confident (read it all; understood it all reasonably well)


Reviewer 3

Summary

This paper presents new safe screening rules for the Sparse-Group Lasso and show a new characterisation of the dual feasible set. Using this they define an efficient algorithm to implement the new screening method using either a sequential or dynamic screening setup. The authors leverage the block-coordinate iterative soft thresholding algorithm (ISTA-BC) to demonstrate the efficiency of their rules in various numerical experiments.

Qualitative Assessment

Safe screening rules are an of key importance in large-scale learning problems and the authors are working on an important problem. My main concern with the paper is that the bottom two plots in Figure 2a both rely on sequential safe rules in some way, while the other methods above do not leverage the sequential aspect. I am concerned that this hides the true performance differences with the top three methods. In particular, no dynamic version of the safe rules is explicitly tested. Similarly, I feel this complicates interpretation of Figures 2b and 3b and weakens the experimental section.

Confidence in this Review

1-Less confident (might not have understood significant parts)


Reviewer 4

Summary

This paper proposed a screening method for regression with group structures. The group level and feature level safe screening rules are introduced and analyzed for theoretical guarantees. It borrows the idea of the SAFE rules for the Lasso. In addition to the theory, the numerical study also support the use of this new screening rule because of the better screening and shorter computation time.

Qualitative Assessment

The proposed method borrows the idea of the SAFE screening rules for the Lasso. I wonder if the authors have compared such SAFE screening methods with sure screening methods (see, for example, Fan and Lv, 2008). $\tau$ seems to be a tuning parameter that balances the variable sparsity and group sparsity. It would be better if the authors explain how to choose a proper $\tau$ in practice and which value is used in the simulation study. In real-data analysis, cross-validation is used to determine $\tau$. Could the authors provide more details to help understand the use of $\tau$ and how sensitive it is to the final estimates. In the simulation study, the authors reported the proportion of active variables. It might be more clear if the authors also report the number of variables eliminated. If the numbers of variables eliminated are more or less equal, the reported proportion makes more sense for the better screening.

Confidence in this Review

2-Confident (read it all; understood it all reasonably well)


Reviewer 5

Summary

This paper proposed the GAP safe screening rules for the Sparse-Group Lasso. To apply the safe rule, the authors give an algorithm to evaluate the dual norm, which they claimed is efficiently.

Qualitative Assessment

Generally speaking, the paper is well written. The general context of the paper is clear. However, there are too many symbols in the Noation part, and all of them are inline formula, which make the paper a little difficult to read. I suggest, for the ease of reading, move some of the formulas to the first place where they appear. For example, move the notations in line 68-70 to line 76. It seems weired to discuss the choice of $\tau$ without the formula of $\Omega_{\tau, w}$ explicitly. In line 24-25, [24] is publised before [11], why the authors said that the work of [24] is following [11]. In line 63, “... the subdifferential ... of the $ \ell_1$_1 norm is $ \sign(\cdot)$”. what is the definition of $ \sign(\cdot)$ in this paper? The subdifferential of the $ \ell_1$_1 norm at 0 the the region $ [-1,1]$. It should be different from the definition of $ \sign(\cdot)$ at 0. In line 160, $\epsilon$-norm is definded as the "unique" nonnegative solution explicitly of the equation. It seems not true. For example, when $\epsilon = 0$, any value which is larger than $\max|x_i|$ would be the solution of the equation. The author should give a strict definition of $\epsilon$-norm.

Confidence in this Review

1-Less confident (might not have understood significant parts)